# Neural activities in music frogs reveal call variations and phylogenetic relationships within the genus *Nidirana*

Ke Fang[1,2,3], Yezhong Tang[1], Baowei Zhang[2] & Guangzhan Fang [1,4✉]

The characteristics of acoustic signals co-evolve with preferences of the auditory sensory system. However, how the brain perceives call variations and whether it can reveal phylogenetic relationships among signalers remains poorly understood. Here, we recorded the neural signals from the Emei music frogs (*Nidirana daunchina*) in response to broadcasted calls of five different species of the same genus. We found that responses in terms of the different amplitudes of various event-related potential (ERP) components were correlated with diversification trends in acoustic signals, as well as phylogenetic relationships between *N. daunchina* and heterospecific callers. Specifically, P2 decreased gradually along the ordinal decline of similarities in acoustic characteristics of calls compared with those from conspecifics. Moreover, P3a amplitudes showed increasing trends in correspondence with callers' genetic distances from the subject species. These observations collectively support the view that neural activities in music frogs can reflect call variations and phylogenetic relationships within the genus *Nidirana*.

[1] Chengdu Institute of Biology, Chinese Academy of Sciences, No.9 Section 4, Renmin Nan Road, 610041 Chengdu, Sichuan, China. [2] School of Life Science, Anhui University, No. 111 Jiulong Road, 230601 Hefei, Anhui, China. [3] Institute of Bio-inspired Structure and Surface Engineering, Nanjing University of Aeronautics and Astronautics, No. 29 Yudao Street, 210016 Nanjing, Jiangsu, China. [4] Key Laboratory of Southwest China Wildlife Resources Conservation (Ministry of Education), China West Normal University, No. 1 Shida Road, 637009 Nanchong, Sichuan, China. ✉email: fanggz@cib.ac.cn

The communication signaling system often plays a critical role in animal speciation for species with predominant acoustic communication, such as insects, anurans, birds, and mammals, as it can potentially create reproductive isolation among closely related species, i.e. those sharing the latest common ancestor[1–4]. Acoustic signals often encode multiple types of information about the signaler's identity, reproductive status, and location[5,6], thus contribute to mate choice, resource defense, species discrimination, and individual recognition[7]. Specifically, since divergence in acoustic traits may play an important role in speciation[8], acoustic signals are well suited to mediate discrimination within and between species[4]. As such, divergence in acoustic traits can be used to predict the pattern of diversification across genera[9].

Previous studies on the coevolution of acoustic signals and sensory systems have shown a strong alignment between signal variations and constraints of sender/receiver physiology[10–12]. For example, there are positive correlations between song complexity and volume of the song control nucleus Hyperstriatum ventrale, pars caudale (HVC) both between and within bird species[11]. Therefore, for females, a complex song can be an index of immediate fitness (the male has a larger brain) and ultimate fitness (he has attractive characteristics that will be passed on to progeny), because the male song is an honest indicator of attributes of a male's brain that could contribute to his fitness or that of his young. Playback and electrophysiological experiments demonstrate that female perception of conspecific signals and discrimination among acoustic signals can be highly precise, even among extensive and complex male vocalizations[11,13]. However, few studies have considered auditory perception and divergence in acoustic signals within a phylogenetic context[14–16]. Accordingly, very little is known about whether dynamic neural activities during auditory perception in the brain can predict the patterns of divergence in acoustic traits and phylogenetic relationships within a clade.

Anurans are not vocal learners; since their vocalizations are relatively fixed, they are traditionally proposed to be genetically determined[17,18]. However, many reports have explored the correlation between the divergence of advertisement calls and geographic distance between populations and/or phylogenetic distance between anuran species, with varying results depending on the species/clades and acoustic parameters. For some clades/acoustic parameters, there is a strong phylogenetic signal[19–23] or correlation with geographic distance[24–27], while for other clades/acoustic parameters this is not the case[28–35]. For example, signal structures in some anuran species appear to be highly conservative within species, while some closely related species often exhibit similar acoustic signals[19–22]. Thus, these anuran vocalizations should be related to genetic differences to a certain extent, consistent with the diversity of anurans acoustic signals[23]. Furthermore, spectral parameters (e.g., frequency modulation and dominant frequency) in anuran vocalizations are strongly influenced by phylogenetic signal[36], and become more conservative under the effects of genetic factors, resulting in correlations with species discrimination[37]. Although a previous study on túngara frogs (Physalaemus pustulosus) showed that female cognition can limit the evolution of sexual signal elaboration[10], very little is known or has been hypothesized about the representation of divergence in acoustic signals and phylogenetic distance within a genus in the auditory central nervous system.

Electroencephalography (EEG) allows a broad canvassing of brain areas composed of multiple neuronal populations that might be involved in different functions including sensory registration, perception, movement, and cognitive processes related to attention, memory, learning, and decision-making[38,39]. Event-related potentials (ERP) are time-locked brain responses to a specific sensory, cognitive, or motor event, whose amplitudes and latencies can be used to examine processing efficiency and time-course of information processing in the brain[40]. In humans, auditory ERPs generally contain three main components (N1, P2, and P3), which peak at latencies of ~80 ms, ~200 ms, and ~300 ms after the stimulus onset, respectively[41,42]. Functionally, N1 is sensitive to selection attention[43]; P2 links to the assessment and classification process for the stimulus, and is sensitive to stimulus complexity and the subject's familiarity with the sound[42]; and P3 is related to psychological constructs of information processing, cognitive processing and memory[44]. Generally, P3 can be divided into two types: P3a and P3b. We focused on P3a, which can be evoked by a novel deviant stimulus using a passive paradigm[45], during which the subjects are not required to engage in any task such as discriminating various acoustic stimuli, thus P3a is the reflection of automatic detection for the novelty in stimuli[46]. In addition, familiar sounds evoke smaller P3a compared with unfamiliar ones[47]. Moreover, for a given component, human-like auditory ERP components with similar stimulus responses and time windows have been found in various taxa such as primates[48], mammals[49,50], and anurans[13,51–55]. Important neuroanatomical features of brain have been conserved during vertebrate brain evolution[56,57]. Moreover, certain aspects of the organization of complex brain networks are highly conserved over different scales and types of measurement across different functional and anatomical networks, and across different species[58]. Hence, similar ERP components across different species may indicate similar brain functions to some extent. For example, recent studies have shown that P2 evoked in the Emei music frog (Nidirana daunchina) possesses human-like time windows[13] and matches behavioral responses based on call identification[59], consistent with the idea that in humans, P2 links to the assessment and classification process for the stimulus. Furthermore, a novel stimulus could evoke greater human-like P3a compared with conspecific calls[13], suggesting P3a elicited in this species is also sensitive to the novelty of the stimuli.

It is traditionally recognized that the genus Nidirana of Ranidae contains five species (N. daunchina, N. adenopleura, N. hainanensis, N. okinavana and N. lini) with a wide allopatrically geographic distribution in east Asia[60,61], although some new species (N. nankunensis, N. yaoica, N. guangdongensis, N. mangveni, N. xiangica and N. leishanensis) have been described in very recent years[62–65]. The Emei music frog (N. daunchina) is a typical seasonal reproductive species that usually produces advertisement calls from within nest burrows in the breeding season[66]. Both behavioral and ERP studies in this species have revealed that the differences in conspecific calls are easily recognized by Emei music frogs on the basis of their different temporal and spectral acoustic characteristics[13,51,67,68]. For example, at the behavioral level, both male and female Emei music frogs respond strongly to conspecific calls, as well as heterospecific calls of N. hainanensis, with a slight response to the calls of N. adenopleura, while no response to the calls of N. lini[67]. The positive correlation between divergences in acoustic signals and phylogenetic distance between the species may cause differences in behavioral response intensity[67]. Accordingly, Emei music frogs may have the ability to perceive phylogenetic distance between species. Electrophysiological studies have shown that different types of conspecific calls can elicit significantly different ERP components, suggesting that ERP components can convey the differences in neural responses to temporal and spectral features of vocalizations[13,51,54,68]. Consequently, it is logical to hypothesize that differences in ERP amplitudes can represent divergences in acoustic signals and phylogenetic relationships within the genus Nidirana when the subjects are exposed to the advertisement calls of the first five confirmed Nidirana species.

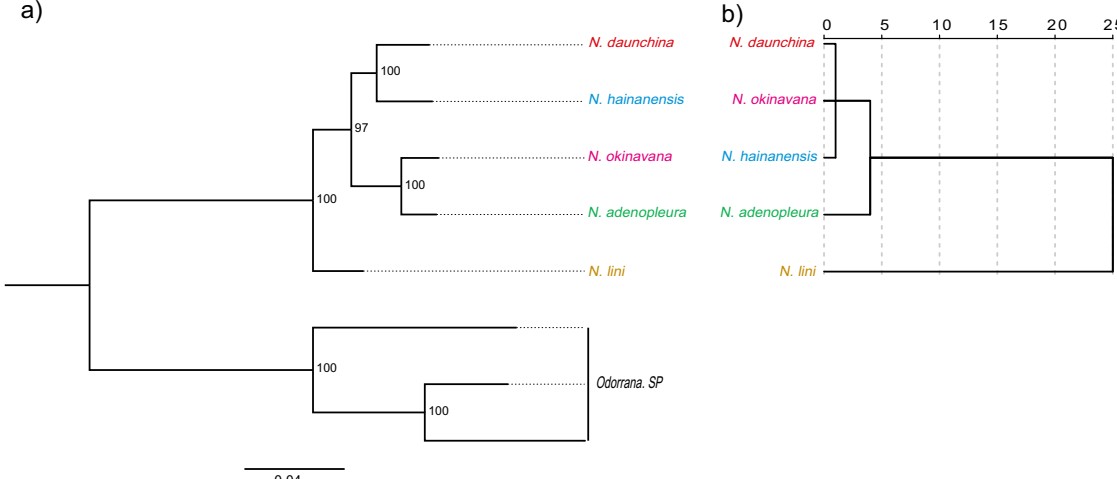

**Fig. 1 Phylogenetic tree and hierarchical cluster of advertisement calls. a** Bayesian inference phylogenetic tree for the five species of genus *Nidirana* based on the 12S and 16S rRNA sequences. Numbers at the branches indicate Bayesian posterior probabilities. **b** Dendrograms illustrating the results of the hierarchical cluster analysis based on the acoustic features of the five *Nidirana* species' calls ($n = 4$ biologically independent samples for each species).

The present study aims to combine acoustic, phylogenetic, and electrophysiological methods to provide additional insight into the evolutionary neuroethology of vocal signaling. To this end, we measured the amplitude and latency of each ERP component in response to the advertisement calls of the five species, from different brain regions of males and females in order to investigate how the auditory nervous system represents signal divergence and phylogenetic relationships in vocal processing. In addition, we analyzed acoustic parameters of the calls and applied Bayesian inference to estimate the evolutionary relationships among the five species, and compared the patterns of auditory perception, signal divergence, and phylogenetic relationship. We predicted that (1) divergence in acoustic signals will align with the phylogenetic distance of the five species, because the former increases with genetic distance in this genus; (2) the amplitudes of the P2 component evoked by the five stimuli will be negatively correlated with divergences in acoustic signals and phylogenetic distances of the five species, since the P2 component is sensitive to the subject's familiarity with the sound, i.e. acoustic similarity relative to conspecific calls in the present study; and (3) the amplitudes of the P3a component evoked by the five stimuli will match the divergence in acoustic signals or the phylogenetic distance of the five species, since the P3a component is sensitive to the novelty in stimuli. The present results showed that P2 amplitudes were negatively correlated with divergences in acoustic characteristics of calls compared with those from conspecifics. Moreover, P3a amplitudes showed increasing trends in correspondence with callers' genetic distances from the subject species. These results collectively support the view that neural activities in music frogs can reflect call variations and phylogenetic relationships within the genus *Nidirana*.

## Results

**Results of the phylogenetic analyses.** For the five species of the genus *Nidirana*, the Bayesian inference analysis showed that *N. daunchina* and *N. hainanensis* formed a distinct clade that was separate from the clade formed by *N. okinavana* and *N. adenopleura*, while *N. Lini* formed the base clade (Fig. 1a). Each node of the tree was strongly supported. Similarly, the results of the genetic distance analysis showed that genetic distance increased gradually from *N. daunchina*, *N. hainanensis*, *N. okinavana*, *N. adenopleura* to *N. lini* (Table 1). These results indicated that *N. daunchina* was more closely related to *N. hainanensis*, *N. okinavana* was more closely related to *N. adenopleura*r, and *N. lini* is closest to the common ancestor of the other four species.

**Acoustic properties of advertisement call of *Nidirana* species.** Acoustic parameters of the advertisement calls of *Nidirana* species are listed in Supplementary Tables 1–6. In order to compare the similarity and divergence of acoustic properties among the advertisement calls of the five *Nidirana* species, temporal and spectral characteristics of the calls were analyzed using hierarchical cluster analysis and multidimensional scaling analysis. The results of the hierarchical cluster analysis revealed three main clusters: the first cluster contained the calls of *N. daunchina*, *N. okinavana* and *N. hainanensis*; the second cluster consisted of the calls of *N. adenopleura* only; and the third cluster consisted of *N. lini* only (Fig. 1b). The multidimensional scaling analysis showed that the discrete distribution pattern of the five *Nidirana* species calls was similar to that from the hierarchical cluster analysis (Supplementary Fig. 1). The degree of similarity between the call of *N. daunchina* and the acoustic properties of the five *Nidirana* species calls exhibited the following trend: *N. daunchina > N. okinavana > N. hainanensis > N. adenopleura > N. lini*. The degree of separation between *N. hainanensis* and *N. daunchina* was significantly greater than that between *N. okinavana* and *N. daunchina* in the two-dimensional solution.

**Amplitude and latency of the N1 component.** The grand average waveforms and difference waveforms are presented in Supplementary Figures 2 and 3, respectively, for each brain area and each acoustic stimulus. For the N1 amplitude evoked in *N. daunchina*, the main effect for the factor "brain area" ($F_{5, 70} = 4.012$, $\varepsilon = 0.840$, partial $\eta^2 = 0.223$, $p = 0.003$) was significant, but not for "sex" ($F_{1, 14} = 0.504$, partial $\eta^2 = 0.035$, $p = 0.489$) or "acoustic stimulus" ($F_{4, 56} = 0.226$, $\varepsilon = 0.715$, partial $\eta^2 = 0.016$, $p = 0.923$). Moreover, there were no significant 2-way and 3-way interactions for the three factors ($p > 0.05$). The N1 amplitude elicited in the left telencephalon (LT) was significantly smaller than that in the left diencephalon (LD) ($p < 0.05$; Supplementary Fig. 4a and Table 2); however, the amplitude in the LT was not significantly different from those of the right telencephalon (RT), right diencephalon (RD) and both sides of the mesencephalon (LM and RM) ($p > 0.05$). The N1 amplitude in RT was significantly smaller than those in LD, RD, LM, and RM ($p < 0.05$; Supplementary Fig. 4a and Table 2). N1 amplitudes

**Table 1 Genetic distance with Tamura three-parameter model between the *Nidirana* and *Odorrana* species based on the 12s,16s gene sequences.**

| ID | Species | 1 | 2 | 3 | 4 | 5 | 6 | 7 |
|----|---------|------|------|------|------|------|------|------|
| 1 | *N. daunchina* | | | | | | | |
| 2 | *N. hainanensis* | 0.0375 | | | | | | |
| 3 | *N. okinavana* | 0.0511 | 0.0531 | | | | | |
| 4 | *N. adenopleura* | 0.0512 | 0.0532 | 0.0257 | | | | |
| 5 | *N. lini* | 0.0513 | 0.0534 | 0.0547 | 0.0535 | | | |
| 6 | *O. exiliversabilis* | 0.1607 | 0.1642 | 0.1632 | 0.1679 | 0.1491 | | |
| 7 | *O. tormotus* | 0.1684 | 0.1818 | 0.1753 | 0.1724 | 0.1592 | 0.0731 | |
| 8 | *O. margaretae* | 0.1571 | 0.1539 | 0.1679 | 0.1696 | 0.1558 | 0.1089 | 0.1260 |

**Table 2 Results of ANOVAs for the amplitudes of N1, P2 and P3a.**

| | $F$ | $\varepsilon$ | $p$ | $\eta^2$ | LSD |
|---|---|---|---|---|---|
| **N1** | | | | | |
| Sex (1,14) | 0.504 | N/A | 0.489 | 0.035 | N/A |
| Acoustic stimulus (4, 56) | 0.226 | 0.715 | 0.923 | 0.016 | N/A |
| Brain area (5, 70) | 4.012 | 0.840 | 0.003* | 0.223 | LD > LT, RT; RD, LM, RM > RT |
| 3-way interaction | 1.028 | 0.379 | 0.428 | 0.068 | N/A |
| **P2** | | | | | |
| Sex (1,14) | 0.001 | N/A | 0.973 | 0.000 | N/A |
| Acoustic stimulus (4,56) | 6.895 | 0.786 | 0.000** | 0.330 | ND, NO > NA, NL |
| Brain area (5, 70) | 2.881 | 0.744 | 0.020* | 0.171 | LD > LT, RD, RM; RT > LT |
| 3-way interaction | 1.318 | 0.328 | 0.166 | 0.086 | N/A |
| **P3a** | | | | | |
| Sex (1,14) | 0.233 | N/A | 0.637 | 0.016 | N/A |
| Acoustic stimulus (4, 56) | 4.811 | 0.754 | 0.002* | 0.256 | NO, NA, NL > ND; NO, NL > NH |
| Brain area (5, 70) | 1.620 | 0.650 | 0.166 | 0.104 | N/A |
| 3-way interaction | 1.349 | 0.321 | 0.148 | 0.088 | N/A |

The symbol '>' denotes that the amplitudes associated with the stimuli or brain areas on the left side of the '>' symbol are significantly greater than those on the right side, and that no significant difference exists among the stimuli or brain areas on the same side of the '>' symbol for each case. Note that nonsignificant 2-way interactions between every two factors are not shown for a better demonstration. The degrees of freedom are shown after each factor.
$F$ is the F-value from ANOVA, $\varepsilon$ the values of epsilon of Greenhouse-Geisser correction, *LSD* least-significant difference test, *LT and RT* the left and right telencephalon, *LD and RD* the left and right diencephalon, *LM and RM* the left and right mesencephalon, *ND* the call of *N. daunchina*, *NH* the call of *N. hainanensis*, *NO* the call of *N. okinavana*, *NA* the call of *N. adenopleura*, *NL* the call of *N. lini*, *N/A* not applicable.
*$p < 0.05$, **$p < 0.001$.

averaged across the six recording sites showed there was no significant difference among the calls of the five species ($p > 0.05$; Fig. 2). In addition, for N1 latency there was no significant main effect and interaction for any factor ($p > 0.05$).

**Amplitude and latency of the P2 component.** For the P2 amplitude elicited in *N. daunchina*, the main effects for the factors "acoustic stimulus" ($F_{4, 56} = 6.895$, $\varepsilon = 0.786$, partial $\eta^2 = 0.330$, $p < 0.001$) and "brain area" ($F_{5, 70} = 2.881$, $\varepsilon = 0.744$, partial $\eta^2 = 0.171$, $p = 0.020$) were significant, but not for "sex" ($F_{1, 14} = 0.001$, partial $\eta^2 = 0.000$, $p = 0.973$). Similarly, there were no significant 2-way and 3-way interactions for the three factors ($p > 0.05$). Multiple comparisons showed that the P2 amplitudes elicited by the calls of *N. daunchina* and *N. okinavana* were significantly larger than those elicited by the calls of *N. adenopleura* and *N. lini* ($p < 0.05$; Fig. 2, Supplementary Fig. 4b and Table 2). However, there were no significant differences in P2 amplitudes between the calls of *N. daunchina* and *N. okinavana*, between the calls of *N. hainanensis* and other species, or between the calls of *N. adenopleura* and *N. lini* ($p > 0.05$). In addition, the P2 amplitude elicited in LD was significantly greater than those in LT, RD and RM, while the P2 amplitude in RT was significantly greater than that in LT ($p < 0.05$; Supplementary Fig. 4b and Table 2). The P2 amplitude in LD was similar to those of RT and LM, and the P2 amplitude in RT was similar to those of LD, RD, LM and RM ($p > 0.05$). Finally, there was no significant main

effect and interaction for P2 latency with respect to any factor ($p > 0.05$).

**Amplitude and latency of the P3a component.** For the P3a amplitude evoked in *N. daunchina*, the main effect for the factor "acoustic stimulus" ($F_{4, 56} = 4.811$, $\varepsilon = 0.754$, partial $\eta^2 = 0.256$, $p = 0.002$) was significant, but not for "sex" ($F_{1, 14} = 0.233$, partial $\eta^2 = 0.016$, $p = 0.637$) or "brain area" ($F_{5, 70} = 1.620$, $\varepsilon = 0.650$, partial $\eta^2 = 0.104$, $p = 0.166$). Moreover, there were no significant 2-way and 3-way interactions for the three factors ($p > 0.05$). The P3a amplitude elicited by the call of *N. daunchina* was significantly smaller than those elicited by the calls of *N. okinavana*, *N. adenopleura* and *N. lini*, while the P3a amplitude evoked by the call of *N. hainanensis* was significantly smaller than those evoked by the calls of *N. okinavana* and *N. lini* ($p < 0.05$, Fig. 2, Supplementary Fig. 4c and Table 2). However, the P3a amplitude evoked by the call of *N. hainanensis* was similar to those evoked by the calls of *N. daunchina* and *N. adenopleura* ($p > 0.05$). Similarly, for P3a latency, there was no significant main effect and interaction for any factor ($p > 0.05$).

**Discussion**
Although some new species have been recently described in the genus *Nidirana*[62–65], it is traditionally recognized that *Nidirana* contains five species (*N. daunchina*, *N. hainanesis*, *N. okinavana*, *N. adenopleura* and *N. lini*) distributed allopatrically in east

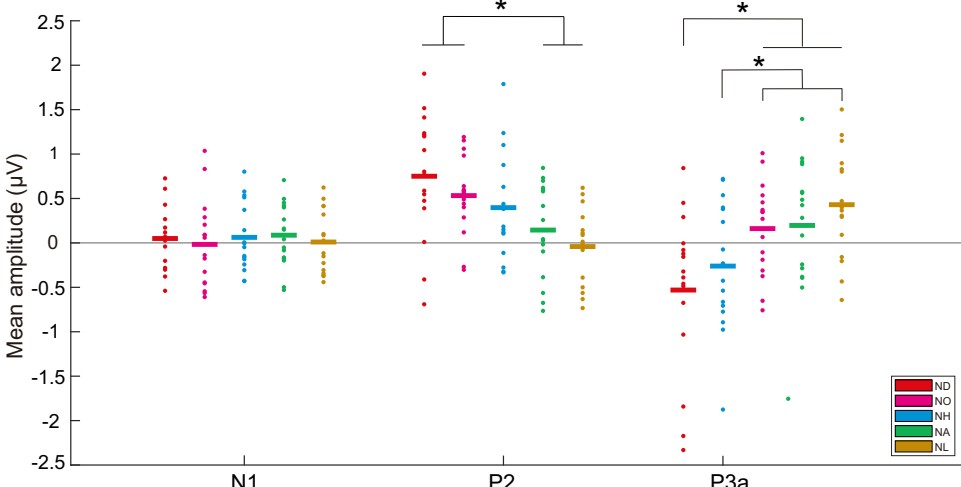

**Fig. 2 Means of N1, P2, and P3a amplitudes in response to different stimulations.** Each asterisk indicates significant differences in amplitudes for a given ERP component between acoustic stimuli ($n = 16$ biologically independent animals; $p < 0.05$; two-way repeated measures ANOVA). Abbreviation: ND, call of *N. daunchina*; NH, call of *N. hainanensis*; NO, call of *N. okinavana*; NA, call of *N. adenopleura*; NL, call of *N. lini*.

Asia[60,61]. Among these species, the Emei music frog (*N. daunchina*) is a relative "recent" species in the genus according to the present results (Fig. 1a) and previously published phylogenies[62–66]. Previous phylogenetic trees have showed somewhat inconsistent topological patterns, likely resulting from the species members and/or models used in the analyses[62,63,65,67]. Thus, we created our own molecular phylogenetic tree using the first five confirmed species in the genus *Nidirana*[60,61] and calculated their genetic distances from each other. The structural profile of our strongly supported phylogenetic tree is topologically the same as three previous trees[62–64] and only slightly different from the other two[65,67]. Here we take the species relationships between the subject and other species in further analyses based on the following considerations. Of the six trees, including ours, four coherently indicate that *N. daunchina* has the closest relationship with *N. hainanesis*, and three support that *N. daunchina* is of distant affiliation with *N. lini*, while *N. okinavana* and *N. adenopleura* are positioned between *N. hainanesis* and *N. lini* regarding relations with *N. daunchina*. Consistent with the topological structure of our phylogenetic tree, the genetic distances between *N. daunchina* and other species increased gradually in the following order: *N. hainanensis*, *N. okinavana*, *N. adenopleura* to *N. lini*.

In order to elucidate the acoustic correlations among the five species, we performed the multidimensional scaling analysis, which unexpectedly showed calls of *N. okinavana*, instead of *N. hainanesis*, being most analogous to those of *N. daunchina*. Moreover, a clustering tree of call similarities was created based on the hierarchical cluster analysis, which is generally comparable with the phylogenetic tree, consistent with the clustering pattern among acoustic signals of the five species, along with genetic divergences within the genus *Nidirana*. In other words, the more closely related species within the same genus exhibit more similar acoustic attributes, suggesting that some call parameters may present a powerful phylogenetic signal in this genus, similar to findings in other anuran species[23,69]. Studies on the concomitant divergence of advertisement calls and genetic characters in anurans have yielded contrasting results, supporting both correspondences between the diverging traits[19–23,69] and a lack of association between the two domains[28–30]. These contrasting results occur because two evolutionary impacts with opposing direction act on the signal system, i.e. phylogeny and adaptation.

The phylogenetic effect trends to keep some inheritable characters unchanged, while the adaptive effect reshapes characters to fit the specific environments. Since the calls of *N. okinavana* – not *N. hainanesis* – are most analogous to those of *N. daunchina*, future studies are needed for a more comprehensive understanding of the combined action of phylogenetic and adaptive impacts on shaping the acoustic attributes of advertisement calls of various species in the genus *Nidirana*.

Our results showed that the calls of *N. daunchina* and *N. okinavana* elicited significantly larger P2 amplitudes in *N. daunchina*, compared with the calls of *N. adenopleura* and *N. lini*. The averaged P2 amplitudes across the six recording sites displayed a responsive pattern with a trend that aligned with the call acoustic cluster, i.e., P2 amplitudes decrease as responding to calls in the following order: *N. daunchina*, *N. okinavana*, *N. hainanesis*, *N. adenopleura* and *N. lini*. P2 is believed to reflect the postsynaptic activities of the neural processes related to some aspects of higher-order perceptual processing such as stimulus evaluation and attention-modulated stimulus classification processes[70,71]. Its amplitude can be enhanced by the subject's familiarity with the acoustic stimulus, or similarity between the target stimulus (the stimulus that the subject should respond to) and the current stimulus[42,72]. Therefore, P2 amplitudes mainly reflect physical signal features found in conspecific calls, and to a lesser extent, signal features of heterospecific calls. In other words, P2 patterns evoked in *N. daunchina* might manifest a neural perceptual process related to processing similarities between conspecific and heterospecific calls. Consistent with this, previous playback experiments also showed that both males and females of *N. daunchina* presented behaviorally strong responses to the heterospecific calls of *N. hainanensis*, a close phylogenetic species to the subject; responses to the calls of *N. adenopleura* were slight, and there were no responses to the calls of *N. lini*[67].

The averaged P3a amplitude in response to the conspecific calls was the smallest compared with those to the heterospecific calls. Importantly, relative P3a amplitudes increased following the order of *N. daunchina*, *N. hainanesis*, *N. okinavana*, *N. adenopleura* and *N. lini*, which is similar to the phylogenetic relations of the genus, while slightly different from the acoustic clustering of the advertisement calls. The presence, magnitude, topography and timing of P3 are often used as metrics of cognitive functions such as classification process in decision-making processes[44–47,73]

and P3a is the reflection of automatic detection for the novelty in stimuli[46], hence familiar sounds would evoke smaller P3a compared with unfamiliar ones[47]. Accordingly, the alignment between P3a amplitudes evoked by the calls of different species and the phylogenetic relations of the genus might result from acoustic similarity relative to conspecific calls, because the more closely related species within the same genus (including the genus *Nidirana*) exhibited more similar acoustic attributes for some anuran species[4,6,36].

N1 is usually associated with the processing of physical stimulus properties, and is strongly modulated by attentional processes[40,43,74,75]. Although there was no significant difference in N1 amplitudes among the five calls from the five species used in the present study, both sides of the diencephalon displayed the highest responsive N1 component to the calls. Since the right hemisphere lateralization exists in auditory attention[76], future research is needed to explore and explain this disagreement.

In summary, our data show that the clustering tree of call similarities is predominantly consistent with the phylogenetic tree for the genus *Nidirana*, indicating that more closely related species exhibit more similar acoustic features in this genus. We demonstrated that divergences in acoustic signals and phylogenetic relationship matched with the hierarchical relationships of P2 and P3a amplitudes respectively, suggesting the coevolution of acoustic signaling and neural processing of communication signals in the Emei music frogs. It is most likely that P2 is a reflection of neural responses to the acoustic features of signals due to the sensitivity of P2 to acoustic similarity relative to conspecific calls. On the other hand, P3a is indicative of neural processes related to decrypting evolutionary history, encoded in acoustic signals of species that show positive correlations between advertisement call variation and phylogenetic distance, since P3a is sensitive to the novelty in stimuli. Future studies measuring these electrophysiological features in other clades with similar positive relationships between call divergence and phylogenetic distance could shed light on whether and how these variables are concomitant in other clades.

## Methods

**Animals**. Sixteen adult Emei music frogs (eight females and eight males) (*N. daunchina*) were captured from the Emei mountain area (29.35°N, 103.17°E, elevation of 1315 m above sea level) of Sichuan, China, during the breeding season. No other species of the genus *Nidirana* could be found in this area. The subjects were separated by sex in opaque plastic tanks (45 × 35 cm, and 30 cm deep) containing about 2 cm depth of water with about 5 cm depth of mud around the bottom walls. The tanks were placed in a room under controlled temperature

conditions (23 ± 1 °C) and relative humidity (70–80%) with a 12:12 light-dark cycle (lights on at 08:00) using a fluorescent lamp. The animals were fed fresh live crickets (bought from a pet food shop) every three days. The subjects measured 4.88 ± 0.29 cm (mean ± SD) in body length and 9.06 ± 2.31 g in body mass at the time of surgery. All studies were conducted under approval of the Animal Care and Use Committee of Chengdu Institute of Biology, Chinese Academy of Sciences (permit number: 20191203).

**Surgery**. The animals were anesthetized by immersion in a 0.1% aqueous solution of tricaine methanesulphonate (MS-222, 100 mg/L); care was taken to minimize their suffering. Seven cortical EEG recording electrodes consisting of miniature stainless-steel screws (0.8 mm in diameter) were implanted into the skull. Six were implanted above the left and right sides of the telencephalon (LT, RT), diencephalon (LD, RD) and mesencephalon (LM, RM) respectively, and the reference electrode was implanted above the cerebellum (Fig. 3). All electrode leads were formvar-insulated nichrome wires with one end tightly secured around the screws and the other end tin soldered to the female-pins of an electrical connector. Electrodes were fixed to the skull with dental acrylic. The connector was covered with a self-sealing membrane (Parafilm® M; Chicago, USA) for waterproofing. Each frog was housed singly in a plastic box (34 × 24.5 × 18.5 cm³) containing water for six days for recovery before conducting further experiments. After all experiments were completed, the subjects were euthanized by overdose of MS-222 solution. The electrode locations were confirmed by injecting hematoxylin dye into the skull hole where the electrodes were previously installed to determine the electrode localizations, for eliminating unexpected data.

**Stimulation**. Three types of stimuli were used in the present study: white noise (WN), conspecific, and heterospecific advertisement calls (Fig. 4 and Supplementary Fig. 5). The duration of WN was equal to the average duration of the calls acquired from five species with 10 ms rise and fall times in sinusoidal periods. The passive Oddball paradigm was applied with WN as the standard stimulus and two calls of different species as the deviant stimuli. For each subject, the experiment contained four sessions, and each of the four stimulus sets was selected randomly and played back during each session. The first stimulus set contained WN, call of *N. daunchina*, call of *N. adenopleura*; the second stimulus set contained WN, call of *N. daunchina*, call of *N. hainanensis*; the third stimulus set contained WN, call of *N. daunchina*, call of *N. lini*; and the fourth stimulus set contained WN, call of *N. daunchina*, call of *N. okinavana* call. Because pseudoreplication may affect the conclusions of statistical analyses in ecological, animal behavior, and neuroscience studies[77,78], its possible effects were controlled by using multiple stimulus exemplars in the present study. To do this, four advertisement calls with almost the same durations were acquired from four different individuals for each species. Each stimulus exemplar was used for four subjects (two males and two females).

In order to compare the similarity and differences of temporal-spectral characteristics among the calls across the five species in the genus *Nidirana*, the following 30 acoustic parameters were measured: call duration, note duration of each of the first five notes, inter-note intervals among the first five notes, both rise time and fall time of each of the first five notes, fundamental frequency for each of the first five notes, and dominant frequency for each of the first five notes (Supplementary Tables 1–6). The twenty temporal parameters were measured from the waveforms of the stimuli using Adobe Audition 3.0 software (San Jose, California, USA), and that thresholding of the vocalizations was set at −48 dB. The ten spectral parameters were measured using Praat software (version 6.1.13) from the spectrograms of the stimuli. These acoustic parameters were selected because

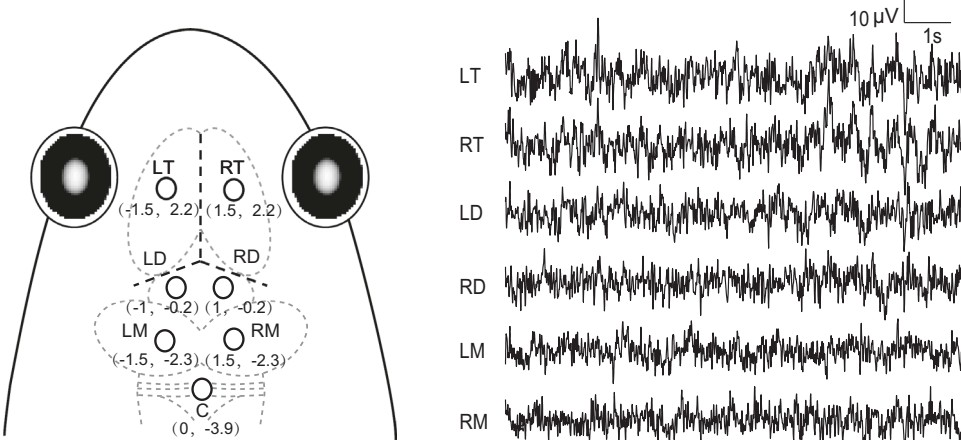

**Fig. 3 Electrode placements and 10 s of typical EEG tracings for each channel.** The intersection of the three dashed bolded lines in the frog head denotes the intersection of suture lines corresponding to lambda. LT, RT, LD, RD, LM, RM denote the left and right sides of the telencephalon, diencephalon, and mesencephalon respectively; while C denotes the reference electrode implanted above the cerebellum.

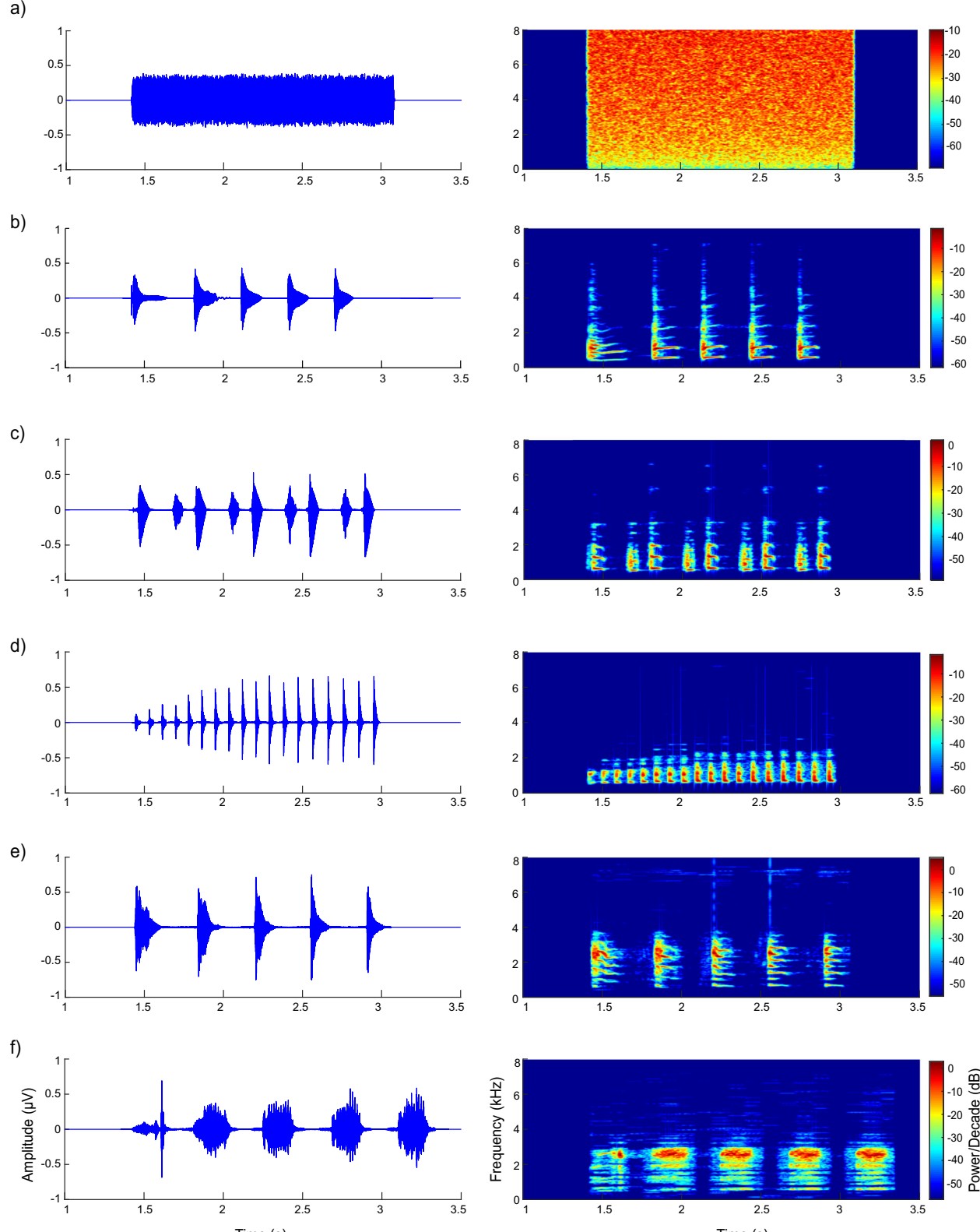

**Fig. 4 Waveforms and spectrograms of the six stimuli. a** White noise, **b** Call of *N. daunchina*, **c** Call of *N. hainanensis*, **d** call of *N. okinavana*, **e** call of *N. adenopleura*, **f** call of *N. lini*.

previous studies have shown that both temporal and spectral characteristics of calls are acoustically suited for enabling species discrimination and individual recognition[79–81]. The values of each acoustic parameter were averaged across the four stimulus exemplars for each species, and were further averaged across the five notes (except for call duration). Then, the hierarchical cluster analysis and multidimensional scaling analysis using Euclidean distance as the metric were used

to determine the similarity and differences of acoustic properties among the stimuli.

**Data acquisition**. An opaque plastic tank (80 × 60 cm² and 60 cm deep) containing mud and water was placed in a soundproof and electromagnetically

shielded chamber (background noise 24.3 ± 0.7 dB). An infrared camera with a motion detector was mounted centrally about one meter above the tank for monitoring the subjects' behaviors. Electrophysiological signals were recorded with a signal acquisition system (OmniPlex64-D, Plexon, USA). After postoperative recovery of six days, the subject was placed in the experimental tank and connected to the signal acquisition system for habituation about 24 h before playback experiments. The band-pass filter was set to 0.05–200 Hz for filtering EEG signals with a sampling rate of 1000 Hz.

The stimuli were played back by two speakers (SME-AFS, Saul Mineroff Electronics, Elmont, NY, USA) placed equidistantly at opposite ends of the experimental tank. Each stimulus was presented through the two speakers simultaneously at 65 dB SPL (re 20 μPa, C-weighting, fast response; Aihua, AWA6291; Hangzhou, China) measured at the center of the tank. For each session, a total of 1,000 stimulus presentations with 1.5 s inter-stimulus interval were broadcasted in a random order, in which WN accounted for 80% of the stimulations while each of the deviant stimuli accounted for 10% of the stimulations. For the deviant stimuli, randomization was constrained to prevent successive presentation of three or more stimuli from the same acoustic category. Each session lasted about 60 min and was divided into three blocks with 5 min break between blocks. Similarly, there was a 5 min break between sessions in order to prevent the animal from becoming fatigued[82]. Before the experiments, a 1000 Hz pure tone was used to calibrate the peak output intensity of each speaker to 70 dB SPL (measured at the center of the tank). All playback orders were randomized using custom-made software in C++. Meanwhile, a trigger pulse was sent to the signal acquisition system at every stimulus onset through the parallel port of a PC for further time-locking analysis.

**Data processing**. To extract ERP components, raw EEG data were filtered by a band-pass filter of 0.25–25 Hz and a notch filter to eliminate possible interference at 50 Hz before averaging the stimulus-locked EEG epochs. EEG signals were then divided into epochs with a duration of 700 ms, including a pre-stimulus baseline of 200 ms. All single EEG epochs were inspected visually and averaged using the EEGLAB toolbox[83] as described in our previous study[13]. For each ERP component, the peak could be found in the grand average waveforms for each brain area and each stimulus. The median was then calculated regardless of brain area or acoustic stimulus. Consequently, the time windows of N1, P2, and P3a were defined as 20–120 ms, 120–220 ms, and 250–350 ms after the onset of stimulation by the median as the midpoint, respectively. Difference waves were acquired by subtracting the ERP amplitudes in response to WN from that of each species' advertisement calls, which could be used to compare relative variations between the ERP responses to the various calls. The amplitude (real number rather than positive number only) and latency of each ERP component were calculated from difference waves within the respective time window by "average amplitude measure" and "half area latency measure", respectively[40]. In addition, the ERP amplitudes and latencies were firstly averaged across four sessions for the calls of *N. daunchina*. Subsequently, the amplitudes and latencies of each ERP component were subjected to further statistical analyses for each brain area and each calls of each species.

**Phylogenetic analyses**. For the phylogenetic analyses, sequences of five *Nidirana* species were used in combination with sequences of three outgroup species, *Odorrana tormotus*, *O. exiliversabilis* and *O. margaretae*[67]. DNA sequence data were obtained from GenBank (Supplementary Table 7), which were then aligned in MEGA X[84] using the Clustal W algorithm with default parameters[85]. The reconstruction of phylogenetic trees was performed with MrBayes version 3.1.2 software[86] using the Bayesian inference method based on the 12 S and 16S rRNA genes. In this process, the chains were sampled every 1000 generations[87,88], using the best-fitting nucleotide substitution model (GTR + I + G) and running the MCMC for 100,000 generations. The first 25% of the total trees were discarded as "burn-in" and the remaining trees were used to generate a majority-rule consensus tree and to calculate Bayesian posterior probabilities. The tree was visualized using FigTree v1.3.1. Genetic distances between *Nidirana* species were calculated in MEGA X using the Tamura three-parameter substitution model[89].

**Statistics and reproducibility**. The normality of distribution and the homogeneity of variances for the ERP values acquired from sixteen animals were estimated with Shapiro-Wilk $W$ test and Levene's test, respectively. The amplitudes and latencies of N1, P2 and P3a components were statistically analyzed using a four-way repeated measured ANOVA with the variables of "stimulus set" (the four stimulus sets), "sex" (female/male), "acoustic stimulus" (the advertisement calls of the five species) and "brain area" (LT, RT, LD, RD, LM and RM). There were no significant main effects for the factor "stimulus set", consistent with the idea that the four stimulus sets were not significantly different at eliciting responses from the subjects. Therefore, all datasets were pooled regardless of "stimulus set" and statistically analyzed using a three-factor repeated measures ANOVA with the other three factors. For each of averaged ERP components across the six recording sites, a two-factor repeated measures ANOVA with the variables of "sex" and "acoustic stimulus" was used. Both main effects and interactions were examined. When significant differences were detected, the data were further analyzed using the least-significant difference test (LSD) for multiple comparisons and simple effects

analysis, respectively. Greenhouse-Geisser $\varepsilon$ values were employed when the assumption of sphericity was violated. The effect size was determined with partial $\eta^2$ (partial $\eta^2 = 0.20$ is set as a small effect size, 0.50 is a medium effect size, and 0.80 is a larger effect size). All analyses were conducted using IBM SPSS Statistics 21.0 (Illinois, USA) with $p < 0.05$ as the significance level.

**Reporting summary**. Further information on research design is available in the Nature Research Reporting Summary linked to this article.

## Data availability
The dataset generated and analyzed in the current study are available at https://doi.org/10.17632/yfnk4zt2hk.1.

## Code availability
All codes used in this work are available upon reasonable request.

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

## Acknowledgements

The authors sincerely thank Di Shen, Jiangyan Shen, Xuemei Xian, Xiaoqin Zhang from Animal Behavior & Bionics Group of Chengdu Institute of Biology, Chinese Academy of Sciences, and Tao Pan, Guiyou Wu from School of Life Science, Anhui University for their discussions and suggestions. This work was supported by grants from the National Natural Science Foundation of China No. 32170504, 31970422, and 31672305 to G.F.

## Author contributions

G.F. conceived the study; G.F., Y.T., and B.Z. designed the research; K.F. performed the animal surgery; K.F. participated in data collection; K.F. carried out data analysis; K.F. drafted the manuscript; Y.T., B.Z., and G.F. revised the manuscript. All authors read and approved the final manuscript.

## Competing interests

The authors declare no competing interests.
