## [Peer Review File · Communications Biology]

Reviewers' comments:

Reviewer #1 (Remarks to the Author):

The authors performed a well-reasoned set of experiments examining the neural responses of *N. dauchina* to conspecific and heterospecific calls. They find correlations between neural responses and call similarity, which are also correlated with genetic similarity.

Major Comments:

1) The argument that call similarity is correlated with genetic distance is valid. The argument that call similarity and events in the ERP is valid. However, it is not a logical step to say that the ERP is involved in processing genetic similarity. This is a classic correlation versus causation argument. The correlations between sound similarity and ERP similarity are expected results, and they may have nothing to do with genetic similarity.

The authors need to temper their discussion regarding these correlations because it undercuts any point that they could make about neural activities and phylogenetic relationships. The only other option is to do another set of experiments with additional control stimuli that account for this.

2) Introduction, Lines 95 -111 (See also Discussion, Lines 305-309): The section starting with "Auditory ERPs generally contain three main components N1, P2, and P3..." must be revised. The authors present a summary of ERPs based on the interpretation of human data with the intent of applying to anuran data. This is not accepted practice for a variety of reasons, including the large differences in morphology and size between humans and frogs. The differences in morphology between humans and frogs will generate differences in the timing and size of ERP recordings which make the two completely incomparable. As far as I know, no one has yet worked out the neural underpinnings of anuran ERPs, but if the authors want to attribute salience to any of the peaks or troughs in the ERP they will need data from a species that at least looks like a frog. We cannot extrapolate from humans to frogs; some folks will bristle at extrapolating from mammals to other mammals.

3) Figure 3B is not effective in communicating what the raw data look like. The reported recordings look like noise. When is the stimulus presented? What are you calling N1, P2, P3? This figure needs to be redone and relabeled to illustrate what these data look like.

4) The text needs a copy editor to check for grammar and syntax. There are minor errors scattered throughout which (at times) diminish the capacity to interpret the paper.

Minor Comments:

Abstract:

The peaks are event-related potentials. They are not biologically salient traits. Saying "It is most likely that P2 is a reflection of advantageously neural reactions (line 38)" is accurate, saying "P3 is involved preferentially in dealing with the biological significances or evolutionary history" (line 40) is not. These peaks are a product of the method of recording. They have a biological underpinning but they do not exist as part of the neural processing of these stimuli, so we must be careful stating that they do or are involved with anything.

Introduction:

Line: 46: I do not understand the first sentence. This needs to be clarified and focused on the topic of the first paragraph.

Line 75: delete "relatively"

Line 76: delete "itself"

Line 77: delete "always" and replace with "often" or something similar.

Line 89: Change "Electroencephalograph" to "Electroencephalography"

Line 94: change "is" to "are"

As a general note, this paper needs a professional copy editor. I will no longer correct grammar, spelling, or syntax.

Methods:

Line 361: Change "30" to "7" in order to match the text and table.

This reviewer cannot comment on the "Phylogenetic Analysis" methodology.

Reviewer #2 (Remarks to the Author):

Dear Authors and Editor,

This is a very interesting manuscript testing the effects of conspecific and heterospecific advertisement calls as neural stimulation for the Emei music frog *Nidirana daunchina*. Results show that different event-related potentials present different trends to the kind of stimulus used. While P2 amplitudes are correlated to the acoustic similarity, P3 amplitudes are correlated to the phylogenetic divergence between species. Authors attribute these results to possible different functions of the brain areas.

In my opinion, this is an interesting and still insufficiently explored subject, and this manuscript brings a meaningful contribution to the field. In my evaluation, the manuscript brings novel and important results, has a solid methodology and is well presented. However, there are suggestions that I would like to point out throughout the text:

Lines 46-47 – I find this sentence confusing and out of place. I suggest rephrasing it or simply excluding

Lines 48-49 – I would not classify insect as "vocal species" due to the different mechanism of sound production. I suggest something such as "species with predominant acoustic communication"

Lines 51 – 52 – "some mammals and most vocal animals" seems strange to me. I suggest excluding the whole sentence and including mammals in the previous sentence as a clade with predominant acoustic communication

Line 62 – Avoid using abbreviations "HVC" without having previously in the text saying what they stand for

Line 73 – Change "genus" to "clade"

Line 77 – Change "always" to something like "most of the times"

Line 75-79 – There are a lot of diverging results for anurans when correlating advertisement call variation to geographic distance between populations and phylogenetic distance between species. Results vary between species/clades and between different acoustic parameters. While for some clades/acoustic parameters there is a strong phylogenetic signal or correlation to geographic distance, for other clades/acoustic parameters this is not true. I suggest including more references, especially more recent ones.

Line 85 – Exclude "in female"

Line 134 – Change "combinate" to "combine"

Line 220 – Change "new" to "recent"

Line 223 – Change "Then" to "Thus,"

Line 227 – Exclude "rationally"

Line 229 – Include a comma "," before and after "including ours"

Line 245-247 – I would not call this a rule, due to the comment I included for lines 75-79

Line 248-249 – I don't think phylogeny is a driving force. I suggest rephrasing

Line 258 – Exclude "very interestingly"

Line 268 – Exclude "our"

Lines 359- 364 – The text says "30 acoustic parameters" and then there is a list of only 7.

That's it. Congratulations and keep up the good work.

Kind Regards,
David Lucas Röhr

Reviewer #3 (Remarks to the Author):

General comments

Fang et al. presents a study combining molecular phylogenetics, bioacoustics, and neurophysiology to determine if attributes of evoked response potentials can be indicative of phylogenetic relationships in a genus of Babina frogs.

Phylogenetic relationships revealed by the tree constructed with a Bayesian inference approach is similar to patterns of similarities in call characteristics as revealed by a hierarchical cluster analysis and a multidimensional scaling analysis. Some of these differences appear to be evident in P2 and P3 amplitudes of evoked potential recordings from *N. daunchina* in response to different heterospecific calls. This study is novel because it identifies correlates of neural activity and divergence patterns of acoustic traits that reflect phylogenetic relationships within a genus of frogs.

Parts of the discussion section should be revised to ensure that the ideas are clear and appropriately conveyed. See specific comments below. I also wonder how generalizable are these findings outside of the genus that the authors have looked at. Can the authors comment on their findings applied to the broader context of animal communication, and better place their results in the context of past work?

Even-related potentials were recorded 6 brain regions in male and female music frogs in response to Babina species. The authors provide a detail analysis and discuss the amplitude data trends in response to calls of different species averaged across recording locations (Fig. 2). However, interaction interactions occur between recording sites and species call types (Fig. S2). that is not well described in the results text.

Specific comments

Line 38: Change "reactions" to "responses".

Lines 46-47: I'm not sure of the idea that you're trying to convey from the opening sentence but it seems to be off.

Lines 49: Change "...it creates reproductive isolation..." to "...it can potentially create reproductive isolation among..."

Line 50: Change "shared" to "sharing"

Line 59: Include "the" between "on" and "coevolution".

Line 60: "constrains" should be "constraints".

Lines 74-75: Move "relatively" to replace "usually".

Line 79: Replace "the phylogeny" with "phylogenetic"

Line 80: Change "component" to "differences".

Line 93: Change "is" to "are".

Line 102: Change "with" to "using a".

Line 106: Delete "In addition" and insert "also" after "can".

Lines 115-118: Put species names in parentheses and move/insert after "...five species". Also, list which species have been described.

Line 124: What does "respond strongly to" refer to? Behavior, neurophysiology, or both?

Line 128: Change "depict" to "convey".

Line 134: Change "combinate" to "combine".

Line 138: Delete "respectively".

Line 139: Insert "the" after "how".

Line 141: Change "conducted" to "applied" and change "analyze" to "estimate".

Line 154: Insert "the" after "of".

Lines 156: The language around describing phylogenetic relationships is unconventional. One can say something like x and y formed a distinct clade that is separate from the clade formed by a, b, and c.

Line 165: Change "were" to "are".

Line 181: Remind the reader that these ERP recordings were conducted on *N. daunchina* so that they don't have to dig for this information in your methods section.

Line 184: Should "amplitude" be plural here because you're referring to amplitudes in response to 5 different call types?

Lines 184-187: When you're making amplitude comparisons here, are they based on absolute values? Is $-1 \mu\text{V}$ not different from $1 \mu\text{V}$? It is stated that amplitudes in LT were significantly smaller than those of LD and RD. This this imply that LT amplitudes were not significantly different from those of RT, LM and RM? Can your results (or supplemental results text) be more thorough in walking the reader through all of these data trends? This comment also applies to describing data relationships depicted in Fig S2 b and c.

Lines 194-195: Move "were significant" to after "brain area..... $p=0.001$)".

Line 219: Insert period after Asia. Capitalize A in "among" and add comma after "them".

Line 220: Is relatively "new" according to the phylogeny in your results? If so, reference your Fig. 1. Is this also based on previously published phylogenies? If so, cite previous papers and change text to "previously published phylogenies".

Line 222: Change "resulted" to "resulting".

Line 223: Delete "Then'.

Line 227: Change "other twos" to "the other two".

Line 230: Delete "ones".

Lines 243-251: This message conveyed in this section of the text is unclear and needs revision. I'm not sure of the authors' intent here and can't recommend specific revisions for clarity.

Line 256: Remind the reader that neural recordings are from *N. dauchina*.

Line 258: Delete "Very".

Line 262: "a specific neural process" is vague. Can you elaborate on this?

Line 265: Please specify what is meant by "target and current stimulus".

Lines 265-268: This text is unclear. Are the authors trying to state that P2 amplitudes mainly reflect physical signal features found in conspecific calls and to a lesser extent, signal features of heterospecific calls?

Line 270: Change "closed" to "close"

Line 281: Insert "in" after "match".

Line 287: Change "At" to "In".

Lines 289-291: The authors need to express care in drawing inferences and comparing cognitive function between humans (as cited in 55-57) and frogs, and attention related to visual (as cited in 55-57) vs auditory tasks.

Line 293: What do you mean by heritability "reliability"? I understand that heritability is defined as

the proportion of the phenotypic variance caused by additive genes, but I'm not sure what you're referring to in using the term reliability.

Lines 299-300: Replace "much comparable" with "consistent". Delete "consistent with".

Line 304: Insert "the" before "co-evolution". Replace "responding" with "processing of communication signals"

Line 313: Gives species name.

Line 344: Should "heterogenous" be "heterospecific"?

Lines 356-357: did exemplars have similar spectra content as well? Data on these exemplars (table of values, plots of frequency spectra) should be provided in supplemental materials.

Lines 359-364: How were call parameters measured? What software was used?

Line 387: Replace "was account" with "accounted for". Replace "probability" with "of stimulation". Make similar changes while describing deviant signals.

Dear Editor and Reviewers,

Thank you for providing us with the opportunity to revise and improve our manuscript. We have considered the comments seriously and have modified the manuscript accordingly as described below. Changes in the text are highlighted in blue.

Sincerely yours,

Ke Fang

Guangzhan Fang

Reviewer #1 (Remarks to the Author):

Question 1. The authors performed a well-reasoned set of experiments examining the neural responses of *N. dauchina* to conspecific and heterospecific calls. They find correlations between neural responses and call similarity, which are also correlated with genetic similarity.

Answer:

Thank you very much for your comments and suggestions.

(Major Comments:)

Question 2. The argument that call similarity is correlated with genetic distance is valid. The argument that call similarity and events in the ERP is valid. However, it is not a logical step to say that the ERP is involved in processing genetic similarity. This is a classic correlation versus causation argument. The correlations between sound similarity and ERP similarity are expected results, and they may have nothing to do with genetic similarity. The authors need to temper their discussion regarding these correlations because it undercuts any point that they could make about neural activities and phylogenetic relationships. The only other option is to do another set of experiments with additional control stimuli that account for this.

Answer:

Yes. It is not suitable to say that the ERP is involved in processing genetic

similarity. For *N. daunchina*, both behavioral and ERP studies have revealed that the differences in acoustic signals are easily recognized by music frogs. A recent study has shown that when the calls of four species (*N. daunchina*, *N. hainanensis*, *N. adenopleura* and *N. lini*) in the genus *Nidirana* were played back, the male and female music frogs respond strongly to the conspecific calls and the heterospecific calls of *N. hainanensis*, with a slight response to the calls of *N. adenopleura*, while no response to the calls of *N. lini* (Cui et al., 2016). The behavioral response intensity is inversely correlated genetic distance shown in the present results, suggesting that the music frogs may have the ability to perceive genetic distance or genetic similarity based on the differences in acoustic signals between species. The perceiving processes could be reflected by the patterns of ERP components. Consequently, we revise the abstract and discussion sections according to ERP component as a reflection of neural responses to specific features encoded in the acoustic signals.

According to “Revised manuscript submission file checklist”, Abstract should be no more than 150 words. Thus, the original sentence in Lines 32-42, “Specifically, P2 amplitudes were the largest when the conspecific calls were...while P3 is involved preferentially in dealing with the biological significances or evolutionary history, encoded in these acoustic signals, because of sensitivity of P3 to the novelty in stimuli.”

has been changed to

“Specifically, P2 decreased gradually along the ordinal decline of similarities in acoustic characteristics of calls compared with those from conspecifics. Moreover, P3a amplitudes showed increasing trends in correspondence with callers’ genetic distances from the subject species. These observations collectively support the view that neural activities in music frogs can reflect call variations and phylogenetic relationships within the genus *Nidirana*.”

The following sentence has been added to Line 126,

“The positive correlation between divergences in acoustic signals and phylogenetic distance between the species may cause differences in behavioral response intensity⁶⁷. Accordingly, Emei music frogs may have the ability to perceive phylogenetic distance

between species.”

The original sentences in Lines 305-309,

“Because the stability for endogenous components is higher than exogenous...”

has been changed to

“It is most likely that P2 is a reflection of neural responses to the acoustic features of signals due to the sensitivity of P2 to acoustic similarity relative to conspecific calls. On the other hand, P3a is indicative of neural processes related to decrypting evolutionary history, encoded in acoustic signals of species that show positive correlations between advertisement call variation and phylogenetic distance, since P3a is sensitive to the novelty in stimuli.”

We delete the sentences in Lines 291-298 in discussion section, “In addition, P3 is an endogenous component of ERP components...with the conserved phylogenetic signals.”

References for this answer (the reference is included in the revised version)

Cui, J. G. et al. Coevolution of male and female response preferences to sexual signals in music frogs. *Asian Herpetol Res* 7, 87-95 (2016).

Question 3. Introduction, Lines 95 -111 (See also Discussion, Lines 305-309): The section starting with “Auditory ERPs generally contain three main components N1, P2, and P3...” must be revised. The authors present a summary of ERPs based on the interpretation of human data with the intent of applying to anuran data. This is not accepted practice for a variety of reasons, including the large differences in morphology and size between humans and frogs. The differences in morphology between humans and frogs will generate differences in the timing and size of ERP recordings which make the two completely incomparable. As far as I know, no one has yet worked out the neural underpinnings of anuran ERPs, but if the authors want to attribute salience to any of the peaks or troughs in the ERP they will need data from a species that at least looks like a frog. We cannot extrapolate from humans to frogs; some folks will bristle at extrapolating from mammals to other mammals.

Answer:

Yes. There are large differences in morphology and size between humans and frogs. And most of studies on auditory ERPs have been conducted in humans to explore various brain functions. Interestingly, compared with human's auditory ERP, similar ERP components (for a given ERP component, with similar time windows, similar responses under similar experimental paradigms and similar biological significance across various species) have been found in monkeys (Arthur and Starr 1984; Swick et al., 1994), cats (Oconnor and Starr 1985), dolphins (Woods et al., 1986), rabbits (Wang et al., 1999), rats (Ehlers et al., 1994) and frogs (e.g., the Emei music frog and the African clawed frog; Fang et al., 2015; Yang et al., 2018; Fan et al., 2018; Fan et al; 2019; Fan et al., 2021).

For example, we have measured ERPs elicited by synthesized white noise (WN), highly sexually attractive (HSA) calls produced by males from inside nests and male calls of low sexual attractiveness (LSA) produced outside of nests, and found that sound classification (discriminating WN and conspecific call) in the music frog occurs within ~100 ms while call discrimination (discriminating HSA and LSA) is accomplished in ~200 ms (Fang et al., 2015). Moreover, WN evoked a larger P3 at ~300 ms than conspecific calls, suggesting the frogs had classified the conspecific calls into one category and perceived WN as novel. These results are consistent with our behavioral study which has showed that male frogs produced fewer overlapping advertisement calls in response to repeating LSA than to HSA playbacks with about a 300 ms delay after stimuli onset (Jiang et al., 2016). This behavioral result is similar to the attention-dependent “voice-specific response” peaking at 320 ms in humans (Levy et al., 2001; 2003). Greenfield (1994) has shown that the effector delay (i.e. the time interval between the trigger from the central nervous system and vocal signal onset) ranges from 50–200 ms in insects. Thus it is reasonable to speculate that males' call timing could be reset by the onset of playbacks and animals could accomplish call identification within around 200 ms.

Interestingly, humans also discriminate auditory objects with a temporal hierarchy: (1) living versus man-made sounds (biotic vs. abiotic sounds) are discriminated first at 70-119 ms, (2) followed by discrimination of human versus

animal nonverbal vocalizations at 169-219 ms, and then (3) continued discrimination of speech as well as subtypes of man-made sounds at 291-357 ms (De Lucia et al., 2010; Murray and Spierer, 2009; Spierer et al., 2011). The findings of similar time windows in frogs and humans, at least for sound classification, suggest that temporal hierarchical mechanisms for auditory perception in classifying various sounds are similar in different species. In addition, it would be expected that strong selective pressures might act on vocal species to identify different conspecific calls/songs as soon as possible for male competition and female choice, which could account for the finding that call discrimination in frogs is faster than human discrimination of human sounds.

Overall, because important neuroanatomical features have been conserved during vertebrate brain evolution (Finlay et al., 2001; Northcutt, 2002), and that certain aspects of the organization of complex brain networks are highly conserved over different scales and types of measurement across different species and for functional and anatomical networks (Bullmore and Sporns 2009), similar ERP components across different species may indicate similar brain functions to some extent. Nevertheless, future studies are needed for a comprehensive understanding of similarities of brain functions reflected by each ERP component to what extent at both behavioral and electrophysiological levels.

We delete the sentences in Lines 106-111,
“In addition, ERP components can be classified as exogenous or endogenous...by the psychological relevance of the stimulus.”

The original sentences in line 111-114,
“Moreover, humanlike auditory ERP components have been found in various taxa including primates, mammals and anurans, indicating similar brain functions because important neuroanatomical features have been conserved during vertebrate brain evolution.”

has been changed to

“Moreover, human-like auditory ERP components with similar time windows and stimulus responses for a given component have been found in various taxa such as

primates⁴⁸, mammals^{49,50}, and anurans^{13,51-55}. Important neuroanatomical features have been conserved during vertebrate brain evolution^{56,57}. Certain aspects of the organization of complex brain networks are highly conserved over different scales and types of measurement across different species, and across different functional and anatomical networks⁵⁸. Hence, similar ERP components across different species may indicate similar brain functions to some extent. For example, recent studies have shown that P2 evoked in the Emei music frog (*Nidirana daunchina*) possesses human-like time windows¹³ and matches behavioral responses based on call identification⁵⁹, consistent with the idea that in humans, P2 links to the assessment and classification process for the stimulus. Furthermore, a novel stimulus could evoke greater human-like P3a compared with conspecific calls¹³, suggesting P3a elicited in this species is also sensitive to the novelty of the stimuli.”

References for this answer (the reference related are included in the revised version)

- Arthur DL, Starr A. Task-relevant late positive component of the auditory event related potential in monkeys resembles P300 in humans. *Science*. 1984; 223:186-8.
- Bernal XE, Akre KL, Baugh AT, Rand AS, Ryan MJ. Female and male behavioral response to advertisement calls of graded complexity in tungara frogs, *Physalaemus pustulosus*. *Behavioral Ecology and Sociobiology*. 2009; 63:1269-79.
- Bullmore E, Sporns O. Complex brain networks: graph theoretical analysis of structural and functional systems. *Nat Rev Neurosci*. 2009; 10:186-98.
- Cui JG, Tang YZ, Narins PM. Real estate ads in Emei music frog vocalizations: female preference for calls emanating from burrows. *Biol Lett*. 2012; 8:337-40.
- Ehlers CL, Kaneko WM, Robledo P, Lopez AL. Long-latency event-related potentials in rats: effects of task and stimulus parameters. *Neuroscience*. 1994; 62:759-69.
- Fang GZ, Jiang F, Yang P, Cui JG, Brauth SE, Tang YZ. Male vocal competition is dynamic and strongly affected by social contexts in music frogs. *Anim Cogn*. 2014; 17:483-94.
- Fang GZ, Yang P, Xue F, Cui JG, Brauth SE, Tang YZ. Sound classification and call discrimination are decoded in order as revealed by event-related potential components in frogs. *Brain Behav Evol*. 2015; 86:232-45.
- Finlay BL, Darlington RB, Nicastro N. Developmental structure in brain evolution. *Behav Brain Sci*. 2001; 24:298-308.
- Greenfield MD, Rand AS. Frogs have rules: selective attention algorithms regulate chorusing in *Physalaemus pustulosus* (Leptodactylidae). *Ethology*. 2000; 106:331-47.
- Jiang, F. et al. Male music frogs compete vocally on the basis of temporal sequence rather than spatial cues of rival calls. *Asian Herpetol Res*. 2015; 6: 305-316.
- Krauzlis RJ, Bogadhi AR, Herman JP, Bollimunta A. Selective attention without a neocortex. *Cortex*. 2018; 102:161-75.
- Northcutt RG. Understanding vertebrate brain evolution. *Integrative and Comparative Biology*.

2002; 42:743-56.

Oconnor TA, Starr A. Intracranial Potentials Correlated with an Event-Related Potential, P300, in the Cat. *Brain Res.* 1985; 339:27-38.

Swick D, Pineda JA, Foote SL. Effects of systemic clonidine on auditory event-related potentials in squirrel-monkeys. *Brain Res Bull.* 1994; 33:79-86.

Wang YP, Kawai Y, Nakashima K. Rabbit P300-like potential depends on cortical muscarinic receptor activation. *Neuroscience.* 1999; 89:423-7.

Woods DL, Ridgway SH, Bullock TH. Middle-and long-latency auditory event-related potentials in dolphins. In: Schusterman JR, Thomas JA, Wood FG, editors. *Dolphin cognition and behavior: a comparative perspective.* London: Lawrence Erlbaum Associates; 1986. p. 61-77

Question 4. (1) Figure 3B is not effective in communicating what the raw data look like. The reported recordings look like noise. When is the stimulus presented? (2) What are you calling N1, P2, P3? This figure needs to be redone and relabeled to illustrate what these data look like.

Answer:

(1) Figure 3B shows spontaneous EEG signals without any artifacts under no stimulus presentation for indicating that surgery procedures are reliable. As it is widely accepted, a brain is considered a chaotic dynamical system and, then, their generated EEG signals are generally chaotic (Sanei and Chambers 2007; Tong and Thankor 2009). In other words, EEG raw data looks like noise.

(2) ERPs are transient changes in the ongoing EEG elicited by sensory, motor, or cognitive events. Because the magnitude of ERPs is often several times smaller than the magnitude of the background EEG (Hu et al. 2010), it is impossible to visually find ERP components from raw EEG signals. Identification of ERP components relies on signal processing methods for enhancing the signal-to-noise ratio (SNR). The most widely used approach to enhance SNR is across-trial averaging in the time domain (Dawson 1951, 1954).

In this revised version, we do not change Figure 3B because this subplot can reflect, to some extent, the qualities of surgical operation and raw EEG signals. We add both grand average waveforms and difference waveforms for different brain regions during playbacks of various acoustic stimuli to the supplemental materials. It should be noted that according to previous studies (McDonald et al., 2005; Ostroff et

al., 2003; Yago et al., 2003), the time windows of N1, P2 and P3 were defined as 30-130 ms, 150-250 ms and 250-350 ms after the onset of stimulation respectively in the original version. In fact, we also used another method to determine the time windows for ERP components: for each ERP component, the peak could be found in the grand average waveforms for each brain area and each stimulus, and the median was then calculated regardless of brain area or stimulus; then, the time window for each ERP component was determined based on the corresponding median and grand average waveforms; consequently, the time windows of N1, P2 and P3 were defined as 20-120 ms, 120-220 ms and 250-350 ms after the onset of stimulation respectively. The results for the two methods are listed in the following table and figure, which show that the significant main effects are same although multiple comparisons show there are some differences between the two methods for brain areas. Because the time windows resulted from the second method match the grand average waveforms better, we report these results in the revised version.

Table 1. Results of ANOVAs for the N1 and P2 amplitudes for the two methods

	F	ε	p	η^2	LSD
N1_M1					
sex (1,14)	0.555	NA	0.469	0.038	N/A
stimulus (4,56)	0.614	0.741	0.655	0.042	N/A
brain area (5, 70)	3.312	0.815	0.010*	0.191	LD, RD>LT; LD, RD, LM>RT
interaction	1.028	0.429	0.429	0.068	N/A
N1_M2					
sex (1,14)	0.504	NA	0.489	0.035	N/A
stimulus (4,56)	0.226	0.715	0.923	0.016	N/A
brain area (5, 70)	4.012	0.840	0.003*	0.223	LD >LT, RT; RD, LM, RM>RT
interaction	1.028	0.379	0.428	0.068	N/A
P2_M1					
sex (1,14)	0.232	NA	0.637	0.016	N/A
stimulus (4,56)	5.494	0.794	0.001*	0.282	ND, NO, NH>NA, NL
brain area (5, 70)	4.841	0.684	0.001*	0.257	LM, RM, LD, RD, RT>LT; LD>RD
interaction	1.217	0.282	0.239	0.080	N/A
P2_M2					
sex (1,14)	0.001	NA	0.973	0.000	N/A
stimulus (4,56)	6.895	0.786	0.000**	0.330	ND, NO >NA, NL
brain area (5, 70)	2.881	0.744	0.020*	0.171	LD >LT, RD, RM; RT>LT
interaction	1.318	0.328	0.166	0.086	N/A

Note: M1, the time windows based on previous studies; M2, the time windows based

on corresponding median and grand average waveforms. LT, RT, LD, RD, LM, RM denote the left and right sides of telencephalon, diencephalon and mesencephalon respectively; N/A, not applicable; ND, the call of *N. daunchina*; NH, the call of *N. hainanensis*; NO, the call of *N. okinavana*; NA, the call of *N. adenopleura*; NL, the call of *N. lini*.

Figure 1. Means and standard errors of N1, P2 and P3a amplitudes in response to different stimulations for the first method (left) and the second method (right). Filled stars indicate that there were significant differences of different amplitudes between acoustic stimuli ($p < 0.05$). Abbreviation: ND, the call of *N. daunchina*; NH, the call of *N. hainanensis*; NO, the call of *N. okinavana*; NA, the call of *N. adenopleura*; NL, the call of *N. lini*.

Because this journal asks all bar graphs should be changed to box-and-whisker or dot-plot format to show data distribution, the original figure 2 has been updated as the following format.

The following sentences have been added to Line 180,

“The grand average waveforms and difference waveforms of each ERP component are presented in Figures S2 and S3, respectively, for each brain area and acoustic stimulus.”

The following sentences have been added to Line 403,

“For each ERP component, the peak could be found in the grand average waveforms for each brain area and each stimulus. The median was then calculated regardless of brain area or acoustic stimulus.”

Reference for this answer (the reference related is included in the revised version)

Dawson GD. A summation technique for detecting small signals in a large irregular background. *J Physiol* 115: 2-3, 1951.

Dawson GD. A summation technique for the detection of small evoked potentials. *Electroencephalogr Clin Neurophysiol* 6: 65–84, 1954.

Hu L, Mouraux A, Hu Y, Iannetti GD. A novel approach for enhancing the signal-to-noise ratio and detecting automatically event-related potentials (ERPs) in single trials. *Neuroimage* 50: 99–111, 2010.

Sanei, S. & Chambers, J. A. EEG Signal Processing (JohnWiley & Son, New York, NY, USA, 2007).

Tong, S. & Thankor, N. V. Quantitative EEG Analysis Methods and Clinical Applications (Artech House, 2009).

McDonald, J. J., Teder-Salejarvi, W. A., Di Russo, F. & Hillyard, S. A. Neural basis of auditory-induced shifts in visual time-order perception. *Nat Neurosci* 8, 1197-1202, doi:10.1038/nn1512 (2005).

Ostroff, J. M., McDonald, K. L., Schneider, B. A. & Alain, C. Aging and the processing of sound duration in human auditory cortex. *Hear Res* 181, 1-7, doi:10.1016/s0378-5955(03)00113-8 (2003).

Yago, E., Escera, C., Alho, K., Giard, M. H. & Serra-Grabulosa, J. M. Spatiotemporal dynamics of the auditory novelty-P3 event-related brain potential. *Cognitive Brain Res* 16, 383-390, doi:10.1016/s0926-6410(03)00052-1 (2003)

Question 5. The text needs a copy editor to check for grammar and syntax. There are minor errors scattered throughout which (at times) diminish the capacity to interpret the paper.

Answer:

We have asked a professional company to proofread this manuscript.

(Minor Comments:)

Question 6. (Abstract) The peaks are event-related potentials. They are not biologically salient traits. Saying “It is most likely that P2 is a reflection of advantageously neural reactions (line 38)” is accurate, saying “P3 is involved preferentially in dealing with the biological significances or evolutionary history” (line 40) is not. These peaks are a product of the method of recording. They have a biological underpinning but they do not exist as part of the neural processing of these stimuli, so we must be careful stating that they do or are involved with anything.

Answer:

Thank you very much for your comments. The original sentence in Lines 32-42, “Specifically, P2 amplitudes were the largest when the conspecific calls were...while P3 is involved preferentially in dealing with the biological significances or evolutionary history, encoded in these acoustic signals, because of sensitivity of P3 to the novelty in stimuli.”

has been changed to

“Specifically, P2 decreased gradually along the ordinal decline of similarities in acoustic characteristics of calls compared with those from conspecifics. Moreover, P3a amplitudes showed increasing trends in correspondence with callers’ genetic distances from the subject species. These observations collectively support the view that neural activities in music frogs can reflect call variations and phylogenetic relationships within the genus *Nidirana*.”

Question 7. (Introduction)

(1) Line: 46: I do not understand the first sentence. This needs to be clarified and focused on the topic of the first paragraph.

(2) Line 75: delete “relatively”

(3) Line 76: delete “itself”

(4) Line 77: delete “always” and replace with “often” or something similar.

(5) Line 89: Change “Electroencephalograph” to “Electroencephalography”

(6) Line 94: change “is” to “are”

(7) As a general note, this paper needs a professional copy editor. I will no longer

correct grammar, spelling, or syntax.

Answer:

- (1) We delete this sentence as Reviewer #2 suggested.
- (2-6) Done.
- (7) We have asked a professional company to proofread this manuscript.

Question 8. (Methods) Line 361: Change “30” to “7” in order to match the text and table. This reviewer cannot comment on the “Phylogenetic Analysis” methodology.

Answer:

In this study, we measured the following 30 acoustic parameters: call duration, note duration of each of the first five notes, inter-note intervals among the first five notes, both rise time and fall time of each of the first five notes, fundamental frequency for each of the first five notes, and dominant frequency for each of the first five notes. The values of each type of parameters were averaged across four exemplars and five notes for each species, thus only 7 acoustic parameters were listed in Table 2. In this revised version, the thirty acoustic parameters are included in Table 2.

The original paragraph in Lines 361-364,
“...the following 30 acoustic parameters were measured: call duration, note duration, fundamental frequency, dominant frequency, inter-note interval, rise time and fall time (all parameters but not call duration were measured for the first five notes of each call, Table 2).”

has been changed to

“...the following 30 acoustic parameters were measured: call duration, note duration of each of the first five notes, inter-note intervals among the first five notes, both rise time and fall time of each of the first five notes, fundamental frequency for each of the first five notes, and dominant frequency for each of the first five notes (Table 2 and Tables S1-S5). The twenty temporal parameters were measured from the waveforms of the stimuli using Adobe Audition 3.0 software (San Jose, California, USA), and that thresholding of the vocalizations was set at -48 dB. The ten spectral

parameters were measured using Praat software (version 6.1.13) from the spectrograms of the stimuli.”

Please also refer to the answer to Question 19 from Reviewer #3.

Reviewer #2 (Remarks to the Author):

Question 1. This is a very interesting manuscript testing the effects of conspecific and heterospecific advertisement calls as neural stimulation for the Emei music frog *Nidirana daunchina*. Results show that different event-related potentials present different trends to the kind of stimulus used. While P2 amplitudes are correlated to the acoustic similarity, P3 amplitudes are correlated to the phylogenetic divergence between species. Authors attribute these results to possible different functions of the brain areas.

Answer:

Thank you very much for your comments and suggestions.

Question 2. (1) In my opinion, this is an interesting and still insufficiently explored subject, and this manuscript brings a meaningful contribution to the field. In my evaluation, the manuscript brings novel and important results, has a solid methodology and is well presented. However, there are suggestions that I would like to point out throughout the text:

(2) Lines 46-47 - I find this sentence confusing and out of place. I suggest rephrasing it or simply excluding

(3) Lines 48-49 - I would not classify insect as “vocal species” due to the different mechanism of sound production. I suggest something such as “species with predominant acoustic communication”

(4) Lines 51-52 - some mammals and most vocal animals’ seems strange to me. I suggest excluding the whole sentence and including mammals in the previous sentence as a clade with predominant acoustic communication

(5) Line 62 - Avoid using abbreviations “HVC” without having previously in the text

saying what they stand for

Answer:

- (1) Thank you very much for your comments and suggestions.
- (2) We delete this sentence.
- (3) “vocal species” has been changed to “species with predominant acoustic communication”.
- (4) Done.
- (5) The original sentence in Line 62, “...of motor song system nucleus HVC both between and within bird species” has been changed to “...of the song control nucleus Hyperstriatum ventrale, pars caudale (HVC) both between and within bird species.”

Question 3.

- (1) Line 73 – Change “genus” to “clade”
- (2) Line 77 – Change “always” to something like “most of the times”
- (3) Line 75-79 – There are a lot of diverging results for anurans when correlating advertisement call variation to geographic distance between populations and phylogenetic distance between species. Results vary between species/clades and between different acoustic parameters. While for some clades/acoustic parameters there is a strong phylogenetic signal or correlation to geographic distance, for other clades/acoustic parameters this is not true. I suggest including more references, especially more recent ones.

Answer:

- (1-2) Done.
- (3) Thank you very much for your suggestions.

For anurans, there are many contrasting results for relationship between advertisement call variation and geographic distances among populations. Clinal variation of advertisement calls related to geographical distance between populations has been reported for different anurans (Velásquez 2014). For example, Annibale and his colleagues found that geographic distance and temperature are the main variables

explaining variation in each of notes of advertisement call of *Dendropsophus nanus* (Annibale et al., 2020). Differences in anuran advertisement calls can be gradual along a geographic gradient (e.g., altitudinal, O'Neill and Beard 2011; latitudinal, Tessarolo et al. 2016; longitudinal, Hasegawa et al. 1999; both latitudinal and altitudinal, Baraquet et al. 2014). Although there are species for which the main differences at a geographical scale reside in variations in temporal rather than in spectral parameters (Andreani et al., 2021; Annibale et al., 2020; Gerhardt and Huber, 2002; Castellano et al., 2002; Velásquez et al., 2013), variation among populations has been found in both spectral and temporal parameters of the advertisement call, and even in sound-producing organs (Lopez et al., 2020). For instance, the subspecies the North American cricket frog (*Acris crepitans*) differ in various parameters of their calls such as call rate, call duration and dominant frequency; the last character presents the most notable differences associated with habitat (Wilczynski and Ryan, 1999). Moreover, for 15 species of monkey tree frogs, only pulse rate is correlated to the geographic distance between populations, while all parameters presented a significant phylogenetic signal (Rohr et al., 2020). Interestingly, pulse duration of the advertisement and aggressive calls of *B. goiana* decreased over the years, probably related to habitat alteration (Andreani et al., 2021). However, there are many studies that have shown no clinal pattern among locations (Gasser et al. 2009; Klymus et al. 2012; Lee et al. 2016; Guerra et al. 2017). For example, no association between acoustic (call duration, pulses/call, pulse duration, interpulse interval, dominant frequency, pulse rate and duty cycle) and linear geographical or altitudinal distances exhibits in *Odontophrynus cordobae* (Grenat et al., 2013).

On the other hand, studies on the concomitant divergence of advertisement calls and genetic characters have also yielded contrasting results, supporting both correspondences between the diverging traits (Andreani et al., 2021; Funk et al., 2009; Velásquez et al., 2013; Sugai et al., 2021) and a lack of association between the two domains (Goutte et al., 2016; Pröhl et al., 2006, 2007). For example, for *Pleurodema thaul*, males emit advertisement calls differing among populations along their distribution in both temporal and spectral structure, and these differences are highly

correlated with genetic variation (Velásquez et al., 2013). In addition, bioacoustic and genetic distances show significant correlations after controlling for geographical distance, suggesting behavioral divergence among populations of *Pleurodema thaul* has a phylogenetic basis (Velásquez et al., 2013). In addition, for the Amazonian frogs *Physalaemus petersi* and *P. freibergi*, genetic divergence was not correlated with geographical distance, rivers or elevation, however, there is a strong positive relationship between genetic divergence and inter-population differences in one call variable, while dominant frequency (Funk et al., 2009). For a tropical anuran assemblage across gradients of environmental heterogeneity in the Pantanal wetlands, acoustic and phylogenetic differences are positively related, while acoustic and body size similarities are negatively related, although to a minor extent (Sugai et al., 2021), suggesting that acoustic partitioning, acoustic adaptation, and allometric constraints play a minor role in shaping the acoustic output of tropical anuran assemblages and that phylogenetic niche conservatism and public information use would influence between-assemblage variation. The North American tree frog *Hyla* group is a genus comprising diploid, tetraploid and triploid hybrid species. These frogs emit a pulsed advertisement call with a pulse rate inversely related to the ploidy level (Foster and Endler, 1999; Gerhardt and Huber, 2002). However, differences in calls between northern and southern túngara frogs can be explained better by geographic distance than by genetic distance, implying significant differences in sexual signals are often not correlated with strong genetic differentiation (Pröhl et al., 2006). Moreover, despite existing morphological constraints, vocalizations of torrent-dwelling species are most probably constrained by the acoustic specificities of torrent habitats and particularly their high level of ambient noise (Goutte et al., 2016). Thus, torrent frogs could thus display similar advertisement calls regardless of phylogenetic relatedness.

Phylogenetic effects, snout to vent length, and water type of breeding habitats have a combined impact on the advertisement calls in anurans (Lin et al., 2020). Hence, divergence of acoustic signals in a geographic scale may result from diverse evolutionary forces acting in parallel. In other words, the combined action of a number of factors (such as natural selection, mutation, genetic drift and hybridization

as well as ecological factors) produces divergence among populations, causing speciation as a potential final outcome.

The original sentences in Line 75-79,

“Anurans are not vocal learners because their vocalizations are usually fixed...genetic drift and shaped by the phylogeny inertia.”

has been changed to

“Anurans are not vocal learners; since their vocalizations are relatively fixed, they are traditionally proposed to be genetically determined^{17,18}. However, many reports have explored the correlation between divergence of advertisement calls and geographic distance between populations and/or phylogenetic distance between anuran species, with varying results depending on the species/clades and acoustic parameters. For some clades/acoustic parameters, there is a strong phylogenetic signal¹⁹⁻²³ or correlation with geographic distance²⁴⁻²⁷, while for other clades/acoustic parameters this is not the case²⁸⁻³⁵. For example, signal structures in some anuran species appear to be highly conservative within species, while some closely related species often exhibit similar acoustic signals¹⁹⁻²².”

Reference for this answer (the reference related is included in the revised version)

- Andreani, T. L., Bastos, R. P., Dias, T. M., Prado, C. P. A. & Morais, A. R. 2021. Acoustic variability among male gladiator frogs, *Boana goiana* (Lutz, 1968) (Anura: Hylidae): an 18-year analysis across several reproductive seasons. *Amphib-Reptilia* **42**, 43-57.
- Annibale, F. S., de Sousa, V. T. T., da Silva, F. R. & Murphy, C. G. 2020. Geographic Variation in the Acoustic Signals of *Dendropsophus nanus* (Boulenger 1889) (Anura: Hylidae). *Herpetologica* **76**, 267-277.
- Baraquet, M., P.R. Grenat, N.E. Salas, and A.L. Martino. 2014. Geographic variation in the advertisement call of *Hypsiboas cordobae* (Anura, Hylidae). *Acta Ethologica* 1-8.
- Funk, W.C., Cannatella, D.C., Ryan, M.J., 2009. Genetic divergence is more tightly related to call variation than landscape features in the Amazonian frogs *Physalaemus petersi* and *P. freibergi*. *J. Evol. Biol.* 22, 1839–1853.
- Foster, S.A., Endler, J.A., 1999. *Geographic Variation in Behavior*. Oxford University Press, Oxford.
- Gasser, H., A. Amezcuita, and W. Hodl. 2009. Who is calling? Intraspecific call variation in the arobobatid frog *Allobates femoralis*. *Ethology* 115:596–607.
- Gerhardt, H., Huber, F., 2002. *Acoustic Communication in Insects and Anurans*. University of Chicago Press, Chicago.
- Guerra, V., A.R. Morais, P.G. Gambale, F.H. Oda, and R.P. Bastos. 2017. Variation of the

- advertisement call of *Physalaemus centralis* Bokermann, 1962 (Anura: Leptodactylidae) in the Cerrado of central Brazil. *Studies on Neotropical Fauna and Environment* 52:103–111.
- Grenat, P. R., Valetti, J. A. & Martino, A. L. 2013. Intra-specific variation in advertisement call of *Odontophrynus cordobae* (Anura, Cycloramphidae): a multilevel and multifactor analysis. *Amphib-Reptilia* 34, 471-482.
- Goutte, S., Dubois, A., Howard, S. D., Marquez, R., Rowley, J. J., Dehling, J. M., ... & Legendre, F. 2016. Environmental constraints and call evolution in torrent - dwelling frogs. *Evolution*, 70(4), 811-826.
- Hasegawa, Y., H. Ueda, and M. Sumida. 1999. Clinal geographic variation in the advertisement call of the wrinkled frog, *Rana rugosa*. *Herpetologica* 55:318–324.
- Klymus, K.E., S.C. Humfeld, and H.C. Gerhardt. 2012. Geographical variation in male advertisement calls and female preference of the wideranging canyon treefrog, *Hyla arenicolor*. *Biological Journal of the Linnean Society* 107:219–232.
- Lee, K.H., P.J.L. Shaner, Y.P. Lin, and S.M. Lin. 2016. Geographic variation in advertisement calls of a Microhylid frog-testing the role of drift and ecology. *Ecology and Evolution* 6:3289–3298.
- Lin, Y. et al. 2020. Advertisement calls of *Fejervarya multistriata* (Anura: Dicroglossidae), with a review of anurans in China. *Anim Biol* 70, 459-481.
- Lopez, C. et al. 2020. Geographic variation in the laryngeal morphology of a widely distributed South-American anuran: behavioural and evolutionary implications. *Zool J Linn Soc* 190, 140-148.
- O'Neill, E.M., and K.H. Beard. 2011. Clinal variation in calls of native and introduced populations of *Eleutherodactylus coqui*. *Copeia* 1:18-28.
- Pröhl H, Koshy R, Mueller U, Rand S, Ryan M. 2006. Geographic variation of genetic and behavioral traits in northern and southern tungara frogs. *Evolution* 60: 1669–1679.
- Pröhl H, Hagemann S, Karsch J, Höbel G. 2007. Geographic variation in male sexual signals in strawberry poison frogs (*Dendrobates pumilio*). *Ethology* 113: 825–837.
- Röhr, D. L., Camurugi, F., Paterno, G. B., Gehara, M., Juncá, F. A., Álvares, G. F., ... & Garda, A. A. 2020. Variability in anuran advertisement call: a multi-level study with 15 species of monkey tree frogs (Anura, Phyllomedusidae). *Canadian Journal of Zoology*, 98(8), 495-504.
- Sugai, L. S. M., Llusia, D., Siqueira, T. & Silva, T. S. F. 2021. Revisiting the drivers of acoustic similarities in tropical anuran assemblages. *Ecology* 102.
- Tessarolo, G., N.M. Maciel, A.R. Morais, and R.P. Bastos. 2016. Geographic variation in advertisement calls among populations of *Dendropsophus cruzi* (Anura: Hylidae). *Herpetological Journal* 26:219-224.
- Velásquez, N. A. 2014. Geographic variation in acoustic communication in anurans and its neuroethological implications. *Journal of Physiology-Paris*, 108(2-3), 167-173.
- Velásquez, N. A., Marambio, J., Brunetti, E., Méndez, M. A., Vásquez, R. A., & Penna, M. 2013. Bioacoustic and genetic divergence in a frog with a wide geographical distribution. *Biological Journal of the Linnean Society*, 110(1), 142-155.
- Wilczynski W, Ryan M. 1999. Geographic variation in animal communication system. In: Foster S, Endler J, eds. *Geographic variation in behavior*. York, NY: New Oxford University Press, 234–261.

Wilkins MR, Seddon N, Safran RJ. 2013. Evolutionary divergence in acoustic signals: causes and consequences. *Trends Ecol Evol* 28: 156–166.

Question 4.

- (1) Line 85 – Exclude “in female”
- (2) Line 134 – Change “combinate” to “combine”
- (3) Line 220 – Change “new” to “recent”
- (4) Line 223 – Change “Then” to “Thus,”
- (5) Line 227 – Exclude “rationally”
- (6) Line 229 – Include a comma “,” before and after “including ours”
- (7) Line 245-247 – I would not call this a rule, due to the comment I included for lines 75-79
- (8) Line 248-249 – I don’t think phylogeny is a driving force. I suggest rephrasing
- (9) Line 258 – Exclude “very interestingly”
- (10) Line 268 – Exclude “our”
- (11) Lines 359-364 – The text says “30 acoustic parameters” and then there is a list of only 7.

Answer:

- (1-6) Done.
- (7) The original sentence in lines 245-247,
“Although for sympatric sibling species a reproductive character displacement can be found, especially in communication signals, while for allopatric related species there seems a rule of signaling as “the more closed species, the more similar features”.”
has been changed to
“Studies on the concomitant divergence of advertisement calls and genetic characters in anurans have yielded contrasting results, supporting both correspondences between the diverging traits^{19-23,69} and a lack of association between the two domains²⁸⁻³⁰.”
- (8) The original sentence in Lines 248-249,

“These happen because two driving forces with reversal direction work on the signal system evolution, i.e. phylogeny and adaptation.”

has been changed to

“These contrasting results occur because two evolutionary impacts with opposing direction act on the signal system, i.e. phylogeny and adaptation. The phylogenetic effect trends to keep some inheritable characters unchanged, while the adaptive effect reshapes characters to fit the specific environments.”

(9-10) Done.

(11) Please refer to the answers to Question 8 from Reviewer #1 and Question 19 from Reviewer #3.

Reviewer #3 (Remarks to the Author):

Question 1. (General comments) Fang et al. presents a study combining molecular phylogenetics, bioacoustics, and neurophysiology to determine if attributes of evoked response potentials can be indicative of phylogenetic relationships in a genus of Babina frogs. Phylogenetic relationships revealed by the tree constructed with a Bayesian inference approach is similar to patterns of similarities in call characteristics as revealed by a hierarchical cluster analysis and a multidimensional scaling analysis. Some of these differences appear to be evident in P2 and P3 amplitudes of evoked potential recordings from *N. daunchina* in response to different heterospecific calls. This study is novel because it identifies correlates of neural activity and divergence patterns of acoustic traits that reflect phylogenetic relationships within a genus of frogs.

Answer:

Thank you very much for your comments and suggestions.

Question 2. Parts of the discussion section should be revised to ensure that the ideas are clear and appropriately conveyed. See specific comments below. I also wonder

how generalizable are these findings outside of the genus that the authors have looked at. Can the authors comment on their findings applied to the broader context of animal communication, and better place their results in the context of past work?

Answer:

Thank you for your suggestions. We revise the discussion section according to your and other reviewers' suggestions.

The original sentence in Lines 285-286,
“N1 is termed ‘sensory’ or ‘exogenous’ component as it is usually associated with the processing of the physical stimulus properties...”

has been changed to

“N1 is usually associated with the processing of physical stimulus properties...”

We delete the sentences in Lines 291-298,
“In addition, P3 is an endogenous component of ERP components...with the conserved phylogenetic signals.”

Important neuroanatomical features have been conserved during vertebrate brain evolution (Finlay et al., 2001; Northcutt, 2002), and that certain aspects of the organization of complex brain networks are highly conserved over different scales and types of measurement across different species and for functional and anatomical networks (Bullmore and Sporns 2009). Thus, it seems reasonable to speculate that similar findings could be found for the genus with positive correlations between advertisement call variation and phylogenetic distance. Thus, the original sentences in Lines 305-309,

“Because the stability for endogenous components is higher than exogenous...”

has been changed to

“It is most likely that P2 is a reflection of neural responses to the acoustic features of signals due to the sensitivity of P2 to acoustic similarity relative to conspecific calls. On the other hand, P3a is indicative of neural processes related to decrypting evolutionary history, encoded in acoustic signals of species that show positive correlations between advertisement call variation and phylogenetic distance, since P3a is sensitive to the novelty in stimuli. Future studies measuring these

electrophysiological features in other clades with similar positive relationships between call divergence and phylogenetic distance could shed light on whether and how these variables are concomitant in other clades.”

In addition, we add related references and discuss the present results in the context of past works.

Question 3. Even-related potentials were recorded 6 brain regions in male and female music frogs in response to Babina species. The authors provide a detail analysis and discuss the amplitude data trends in response to calls of different species averaged across recording locations (Fig. 2). However, interaction interactions occur between recording sites and species call types (Fig. S2). that is not well described in the results text.

Answer:

There is no significant 2-way and 3-way interactions for the factors, and only the statistics for 3-way interaction are shown in Table 3.

The following sentence has been added to Lines 184 and 206,
“Moreover, there were no significant 2-way and 3-way interactions for the three factors ($p>0.05$).”

The following sentence has been added to Line 195,
“Similarly, there were no significant 2-way and 3-way interactions for the three factors ($p>0.05$).”

The word “interaction” in Table 3 has been changed to “3-way interaction”.

The following sentence has been added to the footnote of Table 3,
“Note that nonsignificant 2-way interactions between every two factors are not shown for a better demonstration.”

Question 4. (Specific comments)

(1) Line 38: Change “reactions” to “responses”.

(2) Lines 46-47: I’m not sure of the idea that you’re trying to convey from the opening sentence but it seems to be off.

- (3) Lines 49: Change “...it creates reproductive isolation...” to “...it can potentially create reproductive isolation among...”
- (4) Line 50: Change “shared” to “sharing”
- (5) Line 59: Include “the” between “on” and “coevolution”.
- (6) Line 60: “constrains” should be “constraints”.
- (7) Lines 74-75: Move “relatively” to replace “usually”.
- (8) Line 79: Replace “the phylogeny” with “phylogenetic”
- (9) Line 80: Change “component” to “differences”.
- (10) Line 93: Change “is” to “are”.
- (11) Line 102: Change “with” to “using a”.
- (12) Line 106: Delete “In addition” and insert “also” after “can”.
- (13) Lines 115-118: Put species names in parentheses and move/insert after “...five species”. Also, list which species have been described.
- (14) Line 124: What does “respond strongly to” refer to? Behavior, neurophysiology, or both?
- (15) Line 128: Change “depict” to “convey”.
- (16) Line 134: Change “combinate” to “combine”.
- (17) Line 138: Delete “respectively”.
- (18) Line 139: Insert “the” after “how”.
- (19) Line 141: Change “conducted” to “applied” and change “analyze” to “estimate”.
- (20) Line 154: Insert “the” after “of”.
- (21) Line 165: Change “were” to “are”.

Answer:

- (1) Done.
- (2) As Reviewer #2 suggested, the original sentence in Lines 46-47 has been deleted.
- (3-13) Done.
- (14) The description “at the behavioral level” has been added to Line 124.
- (15-21) Done.

Question 5. Lines 156: The language around describing phylogenetic relationships is unconventional. One can say something like x and y formed a distinct clade that is separate from the clade formed by a, b, and c.

Answer:

The original sentence in Lines 155-158,
“For the five species of genus *Nidirana*, the Bayesian inference analysis showed that the species of *N. daunchina* and *N. hainanensis* were gathered in one branch, the species of *N. okinavana* and *N. adenopleura* were gathered in another branch, while the species *N. Lini* occupied a branch solely (Figure 1A).”

has been changed to

“For the five species of the genus *Nidirana*, the Bayesian inference analysis showed that *N. daunchina* and *N. hainanensis* formed a distinct clade that was separate from the clade formed by *N. okinavana* and *N. adenopleura*, while *N. Lini* formed the base clade (Figure 1A).”

Question 6. Line 181: Remind the reader that these ERP recordings were conducted on *N. daunchina* so that they don't have to dig for this information in your methods section.

Answer:

Thank you very much for your suggestion.

The original sentence in Line 181,
“For the N1 amplitude,...”

has been changed to

“For the N1 amplitude evoked in *N. daunchina*,...”

The original sentence in Line 192,
“For the P2 amplitude,...”

has been changed to

“For the P2 amplitude elicited in *N. daunchina*,...”

The original sentence in Line 203,
“For the P3a amplitude,...”

has been changed to

“For the P3a amplitude evoked in *N. daunchina*,...”

Question 7. Line 184: Should “amplitude” be plural here because you’re referring to amplitudes in response to 5 different call types?

Answer:

Yes. Done.

Question 8. Lines 184-187: (1) When you’re making amplitude comparisons here, are they based on absolute values? Is -1 V not different from 1 V? (2) It is stated that amplitudes in LT were significantly smaller than those of LD and RD. This imply that LT amplitudes were not significantly different from those of RT, LM and RM? Can your results (or supplemental results text) be more thorough in walking the reader through all of these data trends? This comment also applies to describing data relationships depicted in Fig S2 b and c.

Answer:

(1) No. The values of amplitudes acquired from difference waveforms, which are conducted for statistical analysis, are real numbers but not absolute values. Traditionally, ERP amplitudes were quantified (scored) by finding the maximum voltage (or minimum voltage for a negative component) within some time period (http://www.medicine.mcgill.ca/physio/vlab/biomed_signals/eeg_erp.htm or <https://erpinfo.org/blog/2018/7/5/mean-versus-peak-amplitude>). In other words, ERP amplitudes may be positive or negative. In the present study, difference waves were acquired by subtracting the ERP amplitudes in response to white noise from that to each species’ advertisement calls, which could be used to compare relative variations between the ERP responses to the various calls. Accordingly, the values of amplitudes are real numbers.

The original paragraph in Lines 361-364,

“The amplitude and latency of each ERP component were calculated by...”

has been changed to

“The amplitude (real number rather than positive number only) and latency of each ERP component were calculated from difference waves within the respective time window by...”

(2) Yes. The amplitudes in LT were significantly smaller than those of LD and RD, implying that the amplitudes in LT were not significantly different from those of RT, LM and RM.

The original sentence in Lines 184-187,

“The N1 amplitude elicited in left telencephalon (LT) was significantly smaller than those in the left diencephalon (LD) and right diencephalon (RD), while the N1 amplitude in right telencephalon (RT) was significantly smaller than those in LD, RD and left mesencephalon (LM) ($p < 0.05$; Figure S2A and Table 3).”

has been changed to

“The N1 amplitudes elicited in the left telencephalon (LT) were significantly smaller than those in the left diencephalon (LD) ($p < 0.05$; Figure S4A and Table 3), however the amplitudes in the LT were not significantly different from those of the right telencephalon (RT), right diencephalon (RD) and both sides of the mesencephalon (LM and RM) ($p > 0.05$). The N1 amplitude in RT was significantly smaller than those in LD, RD, LM and RM ($p < 0.05$; Figure S4A and Table 3).”

The following sentence has been added to Line 197,

“However, there were no significant differences in P2 amplitudes between the calls of *N. daunchina* and *N. okinavana*, between the calls of *N. hainanensis* and other species, or between the calls of *N. adenopleura* and *N. lini* ($p > 0.05$).”

The following sentence has been added to Line 200,

“The P2 amplitudes in LD were similar to those of RT and LM, and the P2 amplitudes in RT were similar to those of LD, RD, LM and RM ($p > 0.05$).”

The following sentence has been added to Line 210,

“However, the P3a amplitude evoked by the call of *N. hainanensis* was similar to those evoked by the calls of *N. daunchina* and *N. adenopleura* ($p > 0.05$).”

Question 8. (1) Lines 194-195: Move “were significant” to after “brain

area...p=0.001)”. (2) Line 219: Insert period after Asia. Capitalize A in “among” and add comma after “them”.

Answer:

Done.

Question 9. Line 220: Is relatively “new” according to the phylogeny in your results? If so, reference your Fig. 1. Is this also based on previously published phylogenies? If so, cite previous papers and change text to “previously published phylogenies”.

Answer:

Yes. This is based on our results and previously published phylogenies. We cite these references and change the original sentence in Lines 219-221, “...among them the Emei music frog (*N. daunchina*) is a relative “new” species in the genus according to the molecular phylogenetic trees.”

to

“Among these species, the Emei music frog (*N. daunchina*) is a relative “recent” species in the genus according to the present results (Figure 1A) and previously published phylogenies⁶²⁻⁶⁶.”

As Reviewer #2 suggested, the word “new” has been change to “recent”.

Question 10.

Line 222: Change “resulted” to “resulting”.

Line 223: Delete “Then”.

Line 227: Change “other twos” to “the other two”.

Line 230: Delete “ones”.

Answer:

Done.

Question 11. Lines 243-251: This message conveyed in this section of the text is unclear and needs revision. I’m not sure of the authors’ intent here and can’t recommend specific revisions for clarity.

Answer:

The original sentences in Lines 243-251,
“...suggesting that the acoustic signal contains genetic information for those unlearned vocal species...The phylogenetic force tends to keep inheritable genotypic/phylogenic characters unchanged while adaptive one to reshape phylogenic characters to fit the stereotypic environments.”

has been changed to

“...suggesting that some call parameters may present a significant phylogenetic signal in this genus, similar to findings in other anuran species^{23,69}. Studies on the concomitant divergence of advertisement calls and genetic characters in anurans have yielded contrasting results, supporting both correspondences between the diverging traits^{19-23,69} and a lack of association between the two domains²⁸⁻³⁰. These contrasting results occur because two evolutionary impacts with opposing direction act on the signal system, i.e. phylogeny and adaptation. The phylogenetic effect tends to keep some inheritable characters unchanged, while the adaptive effect reshapes characters to fit the specific environments.”

Question 12. Line 256: Remind the reader that neural recordings are from *N. daunchina*.

Answer:

The original sentence in Line 256,
“The current results showed that P2 amplitudes elicited by the calls of *N. daunchina*...”

has been changed to

“Our results showed that the calls of *N. daunchina* and *N. okinavana* elicited significantly larger P2 amplitudes in *N. daunchina*, compared with the calls of *N. adenopleura* and *N. lini*.”

Question 12.

(1) Line 258: Delete “Very”.

(2) Line 262: “a specific neural process” is vague. Can you elaborate on this?

(3) Line 265: Please specify what is meant by “target and current stimulus”.

Answer:

(1) Done.

(2) The original sentence in Line 262,

“P2 is believed to reflect the postsynaptic activities of a specific neural process and some aspects of higher-order perceptual processing such as...”

has been changed to

“P2 is believed to reflect the postsynaptic activities of the neural processes related to some aspects of higher-order perceptual processing such as...”

(3) In lots of experiments, many stimuli will be presented, among which the subjects are asked to select the one to respond. This selected stimulus is known as target stimulus. In other words, a target stimulus is the stimulus that the subject in a test must respond to or attention to.

The original sentence in Lines 264-265,

“...and its amplitude can be enhanced by familiarity or similarity between the target and current stimulus.”

has been changed to

“Its amplitude can be enhanced by the subject’s familiarity with the acoustic stimulus, or similarity between the target stimulus (the stimulus that the subject should respond to) and the current stimulus.”

Question 13. Lines 265-268: This text is unclear. Are the authors trying to state that P2 amplitudes mainly reflect physical signal features found in conspecific calls and to a lesser extent, signal features of heterospecific calls?

Answer:

Yes. The original sentence in Lines 265-268,

“Therefore, P2 patterns evoked in *N. daunchina* might manifest such a neural perceptual process that works mainly on physical aspects of sound signals, such as similarities between conspecific and heterospecific calls.”

has been changed to

“Its amplitude can be enhanced by the subject’s familiarity with the acoustic stimulus, or similarity between the target stimulus (the stimulus that the subject should respond to) and the current stimulus^{42,72}. Therefore, P2 amplitudes mainly reflect physical signal features found in conspecific calls, and to a lesser extent, signal features of heterospecific calls. In other words, P2 patterns evoked in *N. daunchina* might manifest a neural perceptual process related to processing similarities between conspecific and heterospecific calls.”

Question 14.

Line 270: Change “closed” to “close”

Line 281: Insert “in” after “match”.

Line 287: Change “At” to “In”.

Answer:

Done.

Question 15. Lines 289-291: The authors need to express care in drawing inferences and comparing cognitive function between humans (as cited in 55-57) and frogs, and attention related to visual (as cited in 55-57) vs auditory tasks.

Answer:

Thank you for your suggestions. In the revised version, we delete the references cited in 55-57 and we cite a new reference that shows the right hemisphere lateralization might exist in auditory attention in the music frog (Xue et al., 2018).

Please also refer to the answer to Question 3 from Reviewer #1.

Reference for this answer (the reference is included in the revised version)

Xue, F. *et al.* Auditory neural networks involved in attention modulation prefer biologically significant sounds and exhibit sexual dimorphism in anurans. *J Exp Biol* **221**, doi:10.1242/jeb.167775 (2018)

Question 16. Line 293: What do you mean by heritability “reliability”? I understand that heritability is defined as the proportion of the phenotypic variance caused by

additive genes, but I'm not sure what you're referring to in using the term reliability.

Answer:

Sorry for this mistake. The words “heritability” and “reliability” have different meanings. The word “reliability” refers to the test-retest reliability for each ERP component. Meta-analyses of twin studies (van Beijsterveldt and van Baal, 2002) estimated P300 amplitude heritability to be 60%, with a more recent twin study (Hall et al., 2006) yielding a somewhat higher estimate of 69%. P300 heritability is also reflected in biologically-related family members, who demonstrate significant inter-family member correlations for P300 measures elicited by both auditory and visual stimuli (Eischen and Polich, 1994; Polich and Bloom, 1999). Reliability of each ERP variable is usually assessed by calculating the intra-class correlations (ICC) between occasion 1 and 2. For P300 amplitude, the average test-retest reliability of 0.62-0.81 has been reported for a period ranging from a few weeks (Fabiani et al., 1987) to 2 years (Segalowitz and Barnes, 1993). The high reliability and heritability of the P300 amplitude supports its use as candidate endophenotypes (Hall et al., 2006).

Because there is no direct relationship between P300 amplitude heritability/reliability and functions of P300 underlined in the present study, we delete the sentences in Lines 291-298, “In addition, P3 is an endogenous component of ERP components...with the conserved phylogenetic signals.”

References for this answer

- Eischen SE, Polich J. P300 from families. *Electroencephogr Clin Neurophysiol* 1994;92:369-72.
- Fabiani, M., Gratton, G., Karis, D., Donchin, E., 1987. Definition, identification, and reliability of measurement of the P300 components of the event related potentials. In: Ackles, D., Jennings, J., Coles, M. (Eds.), *Advances in Psychophysiology*. JAI Press, New York, pp. 1-78.
- Hall MH, Schulze K, Rijdsdijk F, Picchioni M, Ettinger U, Bramon E, et al. Heritability and reliability of P300, P50 and duration mismatch negativity. *Behav Genet* 2006;36:845-57.
- Polich J, Bloom FE. P300, alcoholism heritability, and stimulus modality. *Alcohol* 1999;17:149-56.
- Segalowitz, S., Barnes, K., 1993. The reliability of ERP components in the auditory oddball paradigm. *Psychophysiology* 30, 451-459.
- van Beijsterveldt CE, van Baal GC. Twin and family studies of the human electroencephalogram: a review and a meta-analysis. *Biol Psychol* 2002;61: 111-38.

Question 17.

Lines 299-300: Replace “much comparable” with “consistent”. Delete “consistent with”.

Line 304: Insert “the” before “co-evolution”. Replace “responding” with “processing of communication signals”

Line 313: Gives species name.

Line 344: Should “heterogenous” be “heterospecific”?

Answer:

Done.

Question 18. Lines 356-357: did exemplars have similar spectra content as well? Data on these exemplars (table of values, plots of frequency spectra) should be provided in supplemental materials.

Answer:

Yes. The four exemplars have similar spectra components for each species. For exemplars acquired from each species, the values of acoustic parameters and plots of frequency spectra are added to the supplemental materials.

Question 19. Lines 359-364: How were call parameters measured? What software was used?

Answer:

We used Praat software (version 6.1.13) to measure fundamental frequency and dominant frequency from the spectrograms of the stimuli, and we used Adobe Audition 3.0 software (San Jose, California, USA) to measure call duration, note duration, inter-note interval, rise time and fall time from the waveforms of the stimuli with thresholding of the vocalizations set as -48 dB.

The original paragraph in Lines 361-364,
“...the following 30 acoustic parameters were measured: call duration, note duration, fundamental frequency, dominant frequency, inter-note interval, rise time and fall time (all parameters but not call duration were measured for the first five notes of each call,

Table 2).”

has been changed to

“...the following 30 acoustic parameters were measured: call duration, note duration of each of the first five notes, inter-note intervals among the first five notes, both rise time and fall time of each of the first five notes, fundamental frequency for each of the first five notes, and dominant frequency for each of the first five notes (Table 2 and Tables S1-S5). The twenty temporal parameters were measured from the waveforms of the stimuli using Adobe Audition 3.0 software (San Jose, California, USA), and that thresholding of the vocalizations was set at -48 dB. The ten spectral parameters were measured using Praat software (version 6.1.13) from the spectrograms of the stimuli.”

Question 20. Line 387: Replace “was account” with “accounted for”. Replace “probability” with “of stimulation”. Make similar changes while describing deviant signals.

Answer:

Done.

REVIEWERS' COMMENTS:

Reviewer #3 (Remarks to the Author):

The authors did a fantastic and thorough job addressing concerns from the three reviewers. I endorse this paper and I have no further concerns.